# Endogenous stimuli-responsive separating microneedles to inhibit hypertrophic scar through remodeling the pathological microenvironment

Zhuo-Ran Yang [1,4], Huinan Suo[2,4], Jing-Wen Fan[3], Niannian Lv[1], Kehan Du[1], Teng Ma[1], Huimin Qin[1], Yan Li[2], Liu Yang[2], Nuoya Zhou[2], Hao Jiang [1] ✉, Juan Tao[2] ✉ & Jintao Zhu [1] ✉

Hypertrophic scar (HS) considerably affects the appearance and causes tissue dysfunction in patients. The low bioavailability of 5-fluorouracil poses a challenge for HS treatment. Here we show a separating microneedle (MN) consisting of photo-crosslinked GelMA and 5-FuA-Pep-MA prodrug in response to high reactive oxygen species (ROS) levels and overexpression of matrix metalloproteinases (MMPs) in the HS pathological microenvironment. In vivo experiments in female mice demonstrate that the retention of MN tips in the tissue provides a slowly sustained drug release manner. Importantly, drug-loaded MNs could remodel the pathological microenvironment of female rabbit ear HS tissues by ROS scavenging and MMPs consumption. Bulk and single cell RNA sequencing analyses confirm that drug-loaded MNs could reverse skin fibrosis through down-regulation of BCL-2-associated death promoter (BAD), insulin-like growth factor 1 receptor (IGF1R) pathways, simultaneously regulate inflammatory response and keratinocyte differentiation via up-regulation of toll-like receptors (TOLL), interleukin-1 receptor (IL1R) and keratinocyte pathways, and promote the interactions between fibroblasts and keratinocytes via ligand-receptor pair of proteoglycans 2 (HSPG2)-dystroglycan 1(DAG1). This study reveals the potential therapeutic mechanism of drug-loaded MNs in HS treatment and presents a broad prospect for clinical application.

Hypertrophic scar (HS) as raised, erythematous, pruritic lesion is mainly caused by the excessive fibroblasts proliferation and disordered collagen deposition following injury to the skin, which seriously affects the appearance, physical and mental health of patients[1,2]. Generally, matrix metalloproteinase 2 (MMP2) and MMP9 are essential enzymes in remodeling extracellular matrix (ECM), which are significantly elevated in the pathological microenvironment of HS[3,4]. Meanwhile, excessive fibroblast proliferation and partially or completely occluded microvessels lead to a hypoxic microenvironment within HS, which remarkably increases reactive oxygen species (ROS)

[1]Hubei Engineering Research Center for Biomaterials and Medical Protective Materials, School of Chemistry and Chemical Engineering, Huazhong University of Science and Technology (HUST), Wuhan 430074, China. [2]Department of Dermatology, Union Hospital, Tongji Medical College, HUST, Wuhan 430022, China. [3]Department of Radiology, Xijing Hospital, The Forth Military Medical University (FMMU), Xi'an 710032, China. [4]These authors contributed equally: Zhuo-Ran Yang, Huinan Suo. ✉e-mail: hustjh@hust.edu.cn; tjhappy@126.com; jtzhu@mail.hust.edu.cn

levels at prolonged low oxygen[5–7]. Therefore, controlled release drug delivery systems in response to endogenous stimuli (MMP2, MMP9 and ROS) to remodel the pathological environment for HS therapy should be promising.

Nowadays, intralesional 5-Fluorouracil (5-Fu) and corticosteroids injections are recommended therapies with different mechanisms in HS treatment[8–11]. Although the use of corticosteroids has been effective for most patients, it has also been associated with troublesome side effects, including hormonophobia, hypopigmentation, subcutaneous fat atrophy, telangiectasias, rebound effects and ineffectiveness[12]. 5-Fu, capable of inhibiting cell proliferation, inducing fibroblast apoptosis, and decreasing collagen production, has been used alone or in combination with corticosteroids to avoid the potential side effects of corticosteroid injections in clinical guideline and practice. Yet, due to the elimination by endogenous dihydropyrimidine dehydrogenase, the half-life of 5-Fu is short, which requires high injection dose and multiple operations[13,14]. Specifically, the high-dose local administration may induce serious side effects (e.g., vasculitis, hyperpigmentation, etythema, purpura, burning sensation), and the invasive injection is accompanied with intense pain[15,16]. Compared to intralesional injection, microneedles (MNs) have attracted significant interests as a new promising transdermal drug delivery platform, due to nonselective loading capacity, minimal invasiveness, simple operation and good biocompatibility[17,18]. Specifically, endogenous stimuli-responsive MNs in response to pathological microenvironment serves as simple and convenient local administration, providing an opportunity for sustainable drug release to enhance the efficacy of drug delivery and minimize the potential side effects[19,20]. Importantly, endogenous stimuli-responsive materials can consume the overexpressed mediators in the pathological microenvironment, which promotes tissue remodeling.

Over the past decade, RNA sequencing (RNA-seq) has become a powerful tool to investigate development and to determine the molecular dysregulation in diseases[21]. Multiple pathways have been identified relevant to the pathogenesis of fibrosis process, including inflammation pathways, anti-apoptosis pathways, growth factor pathways, angiogenesis pathways, etc[22–25]. Among them, anti-apoptosis pathways and growth factor pathways are key pathways of fibroblasts, which are essential to convert an initial stimulus to the development of fibrosis. BCL-2 related proteins are key regulators of mitochondrial apoptosis and have long been identified for its role in fibrosis[26]. Insulin-like growth factor (IGF) is a member of the insulin family, which acts as an anabolic factor to promote cell proliferation and migration. IGF is highly expressed in various fibrotic diseases as well as hypertrophic scars[27]. Nevertheless, the unique therapeutic mechanism of pathological microenvironment-responsive biomaterials on HS treatments is usually neglected in biomaterial science, which is very important for the following basic research and clinical translation.

In this study, we develop a separating MN drug delivery system consisting of gelatin methacryloyl (GelMA) and 5-Fluorouracil acetic acid (5-FuA) prodrug in response to endogenous stimuli (MMP2, MMP9 and ROS) to remodel the pathological microenvironment for HS treatment (Fig. 1). 5-FuA prodrug is designed as ROS-responsive tetrapeptide (PPPK) with 5-FuA at the N-terminus and methacryloyl on the lysine residue, which is UV-crosslinked with GelMA in the tips of separating MN patches. The crosslinking degree can be controlled by UV light irradiation to endow the MN tips with tunable mechanical properties and drug release behaviors[28–32]. Meanwhile, MN base layer can be easily peeled off, leaving MN tips in HS lesions for long-term sustained drug release. In vivo experiments and bulk RNA-seq analysis demonstrate that the separating MN patches can significantly reduce abnormal fibroblast proliferation and collagen fiber deposition through down-regulation of BCL-2-associated death promoter (BAD), insulin-like growth factor 1 receptor (IGF1R) pathways and collagen fibril organization process, simultaneously regulating the inflammatory response and mediating keratinocyte differentiation via up-regulation of toll-like

receptors (TOLL), interleukin-1 receptor (IL1R) and keratinocyte pathways. More importantly, single cell RNA sequencing (scRNA-seq) analysis further investigates the keratinocyte pathways, and demonstrates that the interactions between fibroblasts and keratinocytes play a central role in treating HS tissue with drug-loaded MNs. Besides, drug-loaded MNs can scavenge ROS and consume MMP2 and MMP9 to remodel the pathological microenvironment of HS, which present a broad prospect for clinical application in HS treatment.

## Result
### Fabrication and characterization of separating MN patches
GelMA and 5-FuA were synthesized according to previous reports[33–35], and verified by electrospray time-of-flight high-resolution mass spectrometry (ESI-HRMS) and proton nuclear magnetic resonance ($^1$H NMR) (Supplementary Figs. 1 and 2). 5-FuA prodrug (5-FuA-Pep-MA) and 5-FuA-Pro-Pro were synthesized by standard solid phase peptide synthesis[36–38], and confirmed by ESI-HRMS and $^1$H NMR (Supplementary Figs. 3 and 4). Meanwhile, fluorescein isothiocyanate (FITC)-labeled peptide (FITC-Pep-MA) was synthesized and characterized by ESI-HRMS (Supplementary Fig. 5).

The separating MN patches were prepared by a step-by-step process (Fig. 1). Microscopy images (Fig. 2a) showed that the obtained MN patches contained $10 \times 10$ array, and all the tips maintained intact and quadrangular pyramid shape. The height and bottom side width of every tip were $650 \pm 15 \,\mu m$ and $220 \pm 8 \,\mu m$, respectively, and the distance between adjacent tips was $500 \pm 14 \,\mu m$. No obvious shrinkage, deformation or tip loss was observed with the extension of the UV-crosslinking time. After insertion into agarose gel, MN tips were absence on the base layer of the separating MN patch (Fig. 2b). When gelatin was added to the mould to construct the base layer, a small amount of gelatin entered the cavity to connect the crosslinked tips (Supplementary Fig. 6). Namely, the base layer and the bottom of MN tips were composed of gelatin, which could be fast dissolved by the skin interstitial fluid after penetrating skin tissues. Specifically, the connection of MN tips with the base layer could be easily broken, leaving the tips in skin tissues, due to the presence of dissolvable gelatin in the bottom of MN tips. In comparison, the integral MN prepared with homogeneous GelMA solution maintained the complete MN structure. This separating strategy simplifies the process of transdermal delivery by retention of the drug-loaded tips in the skin for a long-term drug administration.

Since HS has a stiffer and denser tissue structure than normal skin tissue, greater mechanical strength (>0.49 N/needle) is required for efficient drug transport across the skin barrier[11]. Accordingly, the mechanical strength of the obtained MN patches upon different UV-crosslinking time was evaluated (Fig. 3a). It was revealed that MN patches with longer UV-crosslinking time required the larger force to achieve the same compression ratio, demonstrating that the crosslinking density would affect the mechanical strength. More importantly, the obtained MN patches possessed sufficient mechanical strength to penetrate the HS cuticle barrier without disruption. Specifically, the fracture force of a single tip with UV irradiation for 30, 45 and 60 s reached 0.80, 0.77, and 0.67 N/needle, respectively.

GelMA-based MNs can absorb the interstitial fluid within the skin, due to their hydrophilic porous structure, which provides the opportunity for substance exchange between MNs and skin[31,39–41]. Interestingly, the swelling ratio of all the MNs was >300% (Fig. 3b). Notably, when the crosslinking time exceeded 45 s, the swelling ratio was not changed any more, indicating that the UV crosslinking of GelMA was nearly completed.

### Efficient drug loading and responsive release performance of MN patches
The drug loading capacity of MNs was linearly dependent on the drug concentration in GelMA precursor solution (Fig. 3c and

Supplementary Fig. 7). In addition, after MNs were subjected to different UV-crosslinking time, the amount of free 5-FuA-Pep-MA determined by high performance liquid chromatography (HPLC) indicated that the crosslinking efficiency of 5-FuA-Pep-MA was positively correlated with the UV-crosslinking time (Fig. 3d). The higher crosslinking efficiency should also be responsible for the greater mechanical strength and milder swelling of MN patches with longer UV-crosslinking time.

Since the pathological microenvironment of HS includes over-expression of MMPs and high level of ROS[3–7], the separating MN patches for controlled drug release in response to endogenous stimuli was performed. Notably, GelMA UV-crosslinked for 45 s was completely degraded after 84 h (Fig. 3e). The scanning electron microscopy (SEM) images showed that, after enzymatic degradation, the tapered corners and MN tips disappeared and became rounded as a whole, validating the protease-mediated enzymatic response (Fig. 3f). Notably, GelMA degradation was also prompted by higher enzyme activity (Supplementary Fig. 8).

The PPP tripeptide sequence of 5-FuA-Pep-MA with ROS-responsive capability provided another important pathological microenvironment response performance for the separating MN platforms[42–44]. The ROS-responsive cleavage rate of 5-FuA-Pep-MA were positively correlated with $H_2O_2$ concentrations and incubation time (Fig. 3g), revealing that the drug release could be modulated in response to the ROS level in pathological microenvironment at different phases of HS. In order to clarify the ROS-responsive mechanism

of 5-FuA-Pep-MA, HPLC-ESI-HRMS was utilized to characterize the released fragments from crosslinked GelMA/5-FuA-Pep-MA hydrogel (Supplementary Fig. 9), indicating that the responsive bond cleavage occurred between the second and third proline from the N-terminus of the PPP tripeptide sequence.

Meanwhile, the cumulative release of 5-FuA from crosslinked GelMA/5-FuA-Pep-MA hydrogel in the presence of MMP9 and $H_2O_2$ was assessed (Fig. 3h). A sustained and approximately linear release of 5-FuA was obtained from the crosslinked GelMA/5-FuA-Pep-MA hydrogels with the UV-crosslinking time of 10 and 45 s. In comparison, the cumulative release of free 5-Fu in GelMA hydrogel crosslinked for 45 s was much higher (75%, Supplementary Fig. 10), indicating the burst drug release. Overall, the GelMA/5-FuA-Pep-MA MN patches could serve as a therapeutic reservoir for sustained and controlled release of 5-FuA in response to the HS pathological microenvironment.

## Penetration capacity and subcutaneous retention of separating MN patches

Porcine skin possesses a similar structure with human skin[45,46], so rhodamine B (RhB)-loaded MN patches were inserted into porcine skin to evaluate the transdermal delivery capacity. To visualize the drug distribution and penetration in local skin, the fluorescence microscopy images was taken perpendicularly to skin surface at different depths using confocal laser scanning microscope (CLSM) (Fig. 4a). HS model in rabbit ear is a highly reliable, accessible, and measurable model for

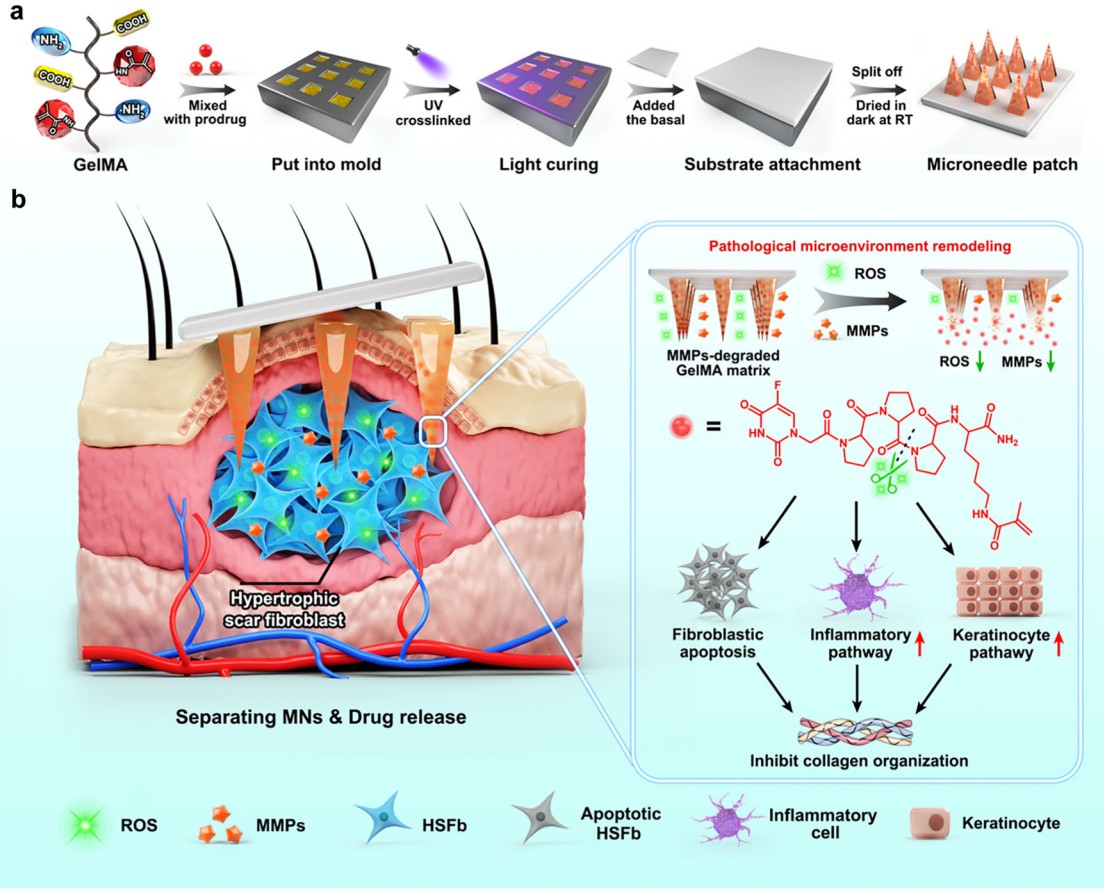

**Fig. 1 | Schematic diagram of the separating MN patch as a pathological microenvironment-responsive peptide prodrug delivery platform. a** Schematic illustration of the fabrication process of a separating MN patch via a step-by-step method. GelMA was mixed with peptide prodrug and photo-initiator to form a precursor solution, poured into MN moulds and crosslinked by UV light. The gelatin solution was then secondarily perfused into the MN mould to form MN base layer. After drying and curing, MN patches were peeled off from the MN mould. **b** Schematic illustration of the drug-loaded MNs on HS treatment. After crosslinked drug-loaded tips penetrate HS tissue, the MN base layer is removed. The slow and sustained drug release from MN tips in response to the HS pathological micro-environment leads to fibroblast apoptosis and down-regulation of collagen fibril organization, simultaneously stimulating the inflammatory response and mediating keratinocyte differentiation to inhibit collagen synthesis of HS tissues.

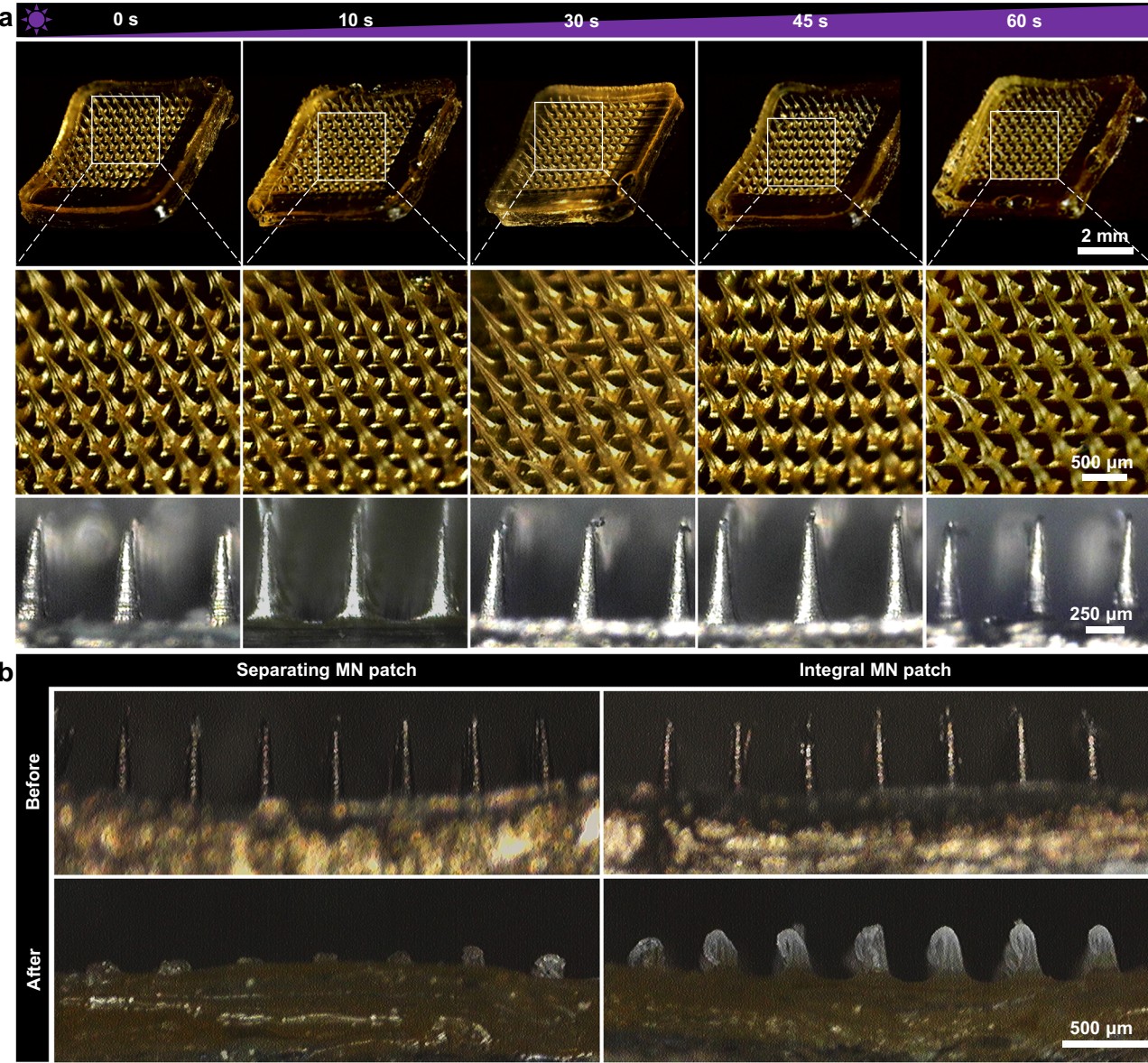

**Fig. 2 | Morphology of MN patches. a** Microscopy images of the obtained MN patches. **b** Microscopy images of the separating and integral MN patches before and after insertion into agarose gel. The scale bar in the last images can be applied to the others in the same panel. Images are representative of *n* = 3 independent experiments.

studying HS[11,15]. Therefore, ex vivo penetration capacity of MNs into HS lesions in rabbit ears was visually assessed by staining the damaged cells with trypan blue. A 10 × 10 array of tip alignment could be observed in the HS tissues treated with MNs (Fig. 4b), confirming that the separating MNs processed enough mechanical strength to break the skin barrier. The hematoxylin and eosin (H&E) histopathology image of the MNs-treated HS tissues presented a deep cavity in the rabbit skin with a depth of ~300 µm, which further confirmed that the separating MNs could pierce through the stratum corneum (Fig. 4c).

Since photochemical crosslinking endowed MN tips with slow degradation activity in the HS pathological microenvironment, in vivo subcutaneous retention of MN tips was assessed in the ventral skin surface of rabbit ear (Fig. 4d). The tip alignment could be observed within 5 days after penetration of a separating MN patch. In comparison, the microhole array disappeared within a few hours, after insertion of an integral MN patch into the HS tissue (Supplementary Fig. 11). The subcutaneous retention of MNs was further verified by in vivo fluorescence microscopy imaging and histological analysis (Supplementary Figs. 12 and 13). Significant fluorescent spot

corresponding to MN tips embedded in the skin could be observed initially, followed by gradual dimming over time. In comparison, when the integral MNs without separating capacity penetrated the skin, no fluorescence signal was observed in the skin of the mice. After MNs penetrated the skin, MN tips were fully embedded in skin tissue, and the sizes of MN tips gradually decreased with time extension. Notably, above results also demonstrated the long-term retention ability of the separating MNs.

To demonstrate the minimal invasion of the separating MNs as a transdermal drug delivery device, five healthy volunteers were recruited to assess the puncturing and recovery of the skin barrier after penetration of separating MNs. Trans epidermal water loss (TEWL), erythem value, melanin value and corneometer value were measured at indicated intervals. After penetration of separating MNs, the mean TEWL value immediately increased from 8.6 to 10.8, followed by gradual decrease to prior level within 60 min (Fig. 4e). Similarly, erythem value increased from 290 to 410 upon penetration and decreased to 300 within 60 min (Fig. 4f). Meanwhile, corneometer value and melanin value showed negligible changes

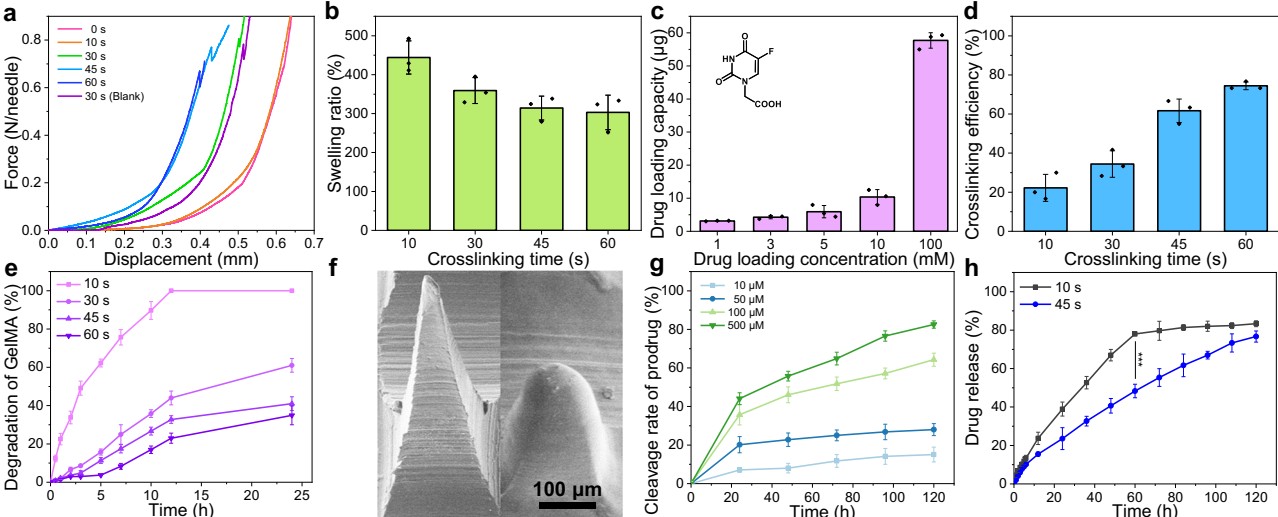

**Fig. 3 | Characterization, efficient drug loading and responsive drug release of MN patches. a** Force curve of MN patches with different crosslinking time. Arrows indicate fracture force. **b** Swelling ratios of MNs with different crosslinking time. **c** Drug loading amount of 5-FuA prepared from various concentrations of drug loading solutions. **d** Crosslinking efficiency of MNs with different crosslinking time. **e** Enzymatic degradation of GelMA with different crosslinking time. **f** SEM images of MN patches before (left) and after (right) enzymatic degradation ($n = 3$ independent experiments). **g** ROS-responsive cleavage of 5-FuA-Pep-MA prodrug incubated with different concentrations of $H_2O_2$. **h** Drug release curves of dual responsiveness drug delivery platform with different crosslinking time. The data in (**b**–**e**), (**g**, **h**) are presented as mean ± SD ($n = 3$ independent experiments). The data were analyzed by $t$-test. $P$-value: ***$P < 0.001$.

(Supplementary Fig. 14). Overall, these results demonstrated that the penetration of separating MNs induced minimal invasion and the punctured skin could be rapidly recovered to its original state.

## In vitro fibroblast cytotoxicity, inhibition of migration activity and scavenging capacity of intracellular ROS

To evaluate the fibroblast cytotoxicity and of 5-Fu and 5-Fu derivatives, cell counting kit-8 (CCK-8) assay and live/dead staining assay were performed, respectively. Specifically, the cell viability was ~57.1% in 5-FuA group compared with 46.9% in 5-Fu group at the concentration of 250 μM, indicating that the inhibition of fibroblast proliferation by 5-Fu was slightly better than 5-FuA ($P < 0.05$, Fig. 5a and d). Notably, the inhibitory effect of 5-FuA, 5-FuA-Pep-MA and 5-FuA-Pro-Pro on fibroblasts showed no significant difference at all the concentrations (ns, Fig. 5b and d). Meanwhile, the inhibition of fibroblast migration by 5-FuA, 5-FuA-Pep-MA and 5-FuA-Pro-Pro was tested by in vitro wound scratch assay. The optical microscopy images showed that almost complete wound closure could be observed within 48 h in the control group (Fig. 5e). In comparison, the quantitative analysis of these images revealed that the wound healing rate in 5-FuA, 5-FuA-Pep-MA and 5-FuA-Pro-Pro groups was only 48.3%, 51.6% and 53.2%, respectively, after 48 h of incubation (Fig. 5c). Therefore, the results demonstrated that 5-FuA-Pep-MA could be potentially used as a therapeutic prodrug to inhibit the fibroblast proliferation and migration for HS treatment. In addition, the in vitro scavenging capacity of intracellular ROS was initially evaluated via a ROS assay kit. As shown in Supplementary Fig. 15, the fluorescence intensity in the experimental groups was significantly reduced with the increased concentration of 5-FuA-Pep-MA, confirming that 5-FuA-Pep-MA could serve as a ROS scavenger to efficiently remove the endogenous ROS in human hypertrophic scar fibroblast (HSFb).

## In vivo therapeutic efficacy of separating MN patches on HS

The HS model in rabbit ear was constructed to investigate in vivo therapeutic efficacy of separating MNs (Fig. 6a). We randomly divided all rabbits into five groups, including NS, HS, drug-free MNs, drug-loaded MNs and 5-Fu injection groups. Figure 6b depicted the photographs of skin lesions in all groups at different time intervals. After the 3rd

treatment and 1-week follow-up, the lesions in the drug-loaded MNs group were smoother, and the scar boundaries almost disappeared. Although a similar therapeutic effect was obtained in 5-Fu injection group, newly formed needle-point scars were observed in the lesions, which could be ascribed to secondary damage by syringe injection. Moreover, drug-free MNs exhibited limited therapeutic efficacy, leading to the decreased height and blurred border of HS lesions. The results revealed that drug-loaded MNs were more effective for HS treatment.

Furthermore, H&E staining and Masson's trichrome staining depicted the changes of the pathological HS lesions before and after different treatments (Fig. 6d, e). The dermis in experimental groups possessed many fibroblasts and thick collagen bundles before administration. After administration with drug-loaded MNs for three times, the thickness of HS tissue dramatically decreased. The magnified images further revealed the obvious decrease of the collagen deposition and the regular arrangement of fibroblasts. In comparison, HS tissue treated with 5-Fu injection displayed a bit thicker dermis and relatively disordered collagen distribution. As for the drug-free MNs group, the thickness of HS tissue showed a slight decline after the 3rd administration. To further verify the therapeutic effect, the scar elevation index (SEI) was calculated based on the images of H&E staining (Fig. 6c). After the 3rd administration, the SEI value of the drug-loaded MNs group decreased from 3.5 to 1.4, which of the HS group remained 3.3 ($P < 0.05$). Meanwhile, the SEI value of 5-Fu injection group was 2.2, which was significantly higher than that of the drug-loaded MNs group ($P < 0.05$). These results demonstrated that the drug-loaded MNs could effectively inhibit fibroblast over-proliferation and collagen synthesis, and exhibited better therapeutic effect than 5-Fu injection.

To further investigate the remodeling of the pathological microenvironment by the drug-loaded MNs, in vivo ROS scavenging capacity was initially evaluated via dihydroethidium (DHE) and 8-hydroxy-2'-deoxyguanosine (8-OHdG) staining[47,48]. As shown in Fig. 6f, the fluorescence intensity in the drug-loaded MNs group was significantly reduced, confirming that drug-loaded MNs could serve as a ROS scavenger to efficiently remove the endogenous ROS. More importantly, the immunohistochemistry and immunofluorescence staining were performed to detect the expression of MMP2 and MMP9 in the pathological

microenvironment of HS (Fig. 6f and Supplementary Fig. 16). Interestingly, both the drug-loaded MNs and drug free MNs could significantly decrease the levels of MMP2 and MMP9, which inhibited the fibroblast migration. Therefore, the results revealed that the drug-loaded MNs could remodel the microenvironment of HS tissues by ROS scavenging

and MMPs consumption. More importantly, drug-loaded MNs stored at room temperature and 50% relative humidity for 9 months were applied for the treatment in rabbit ear HS model, which still had excellent biological effects after long-term storage, as well as good therapeutic effects on HS (Supplementary Fig. 17).

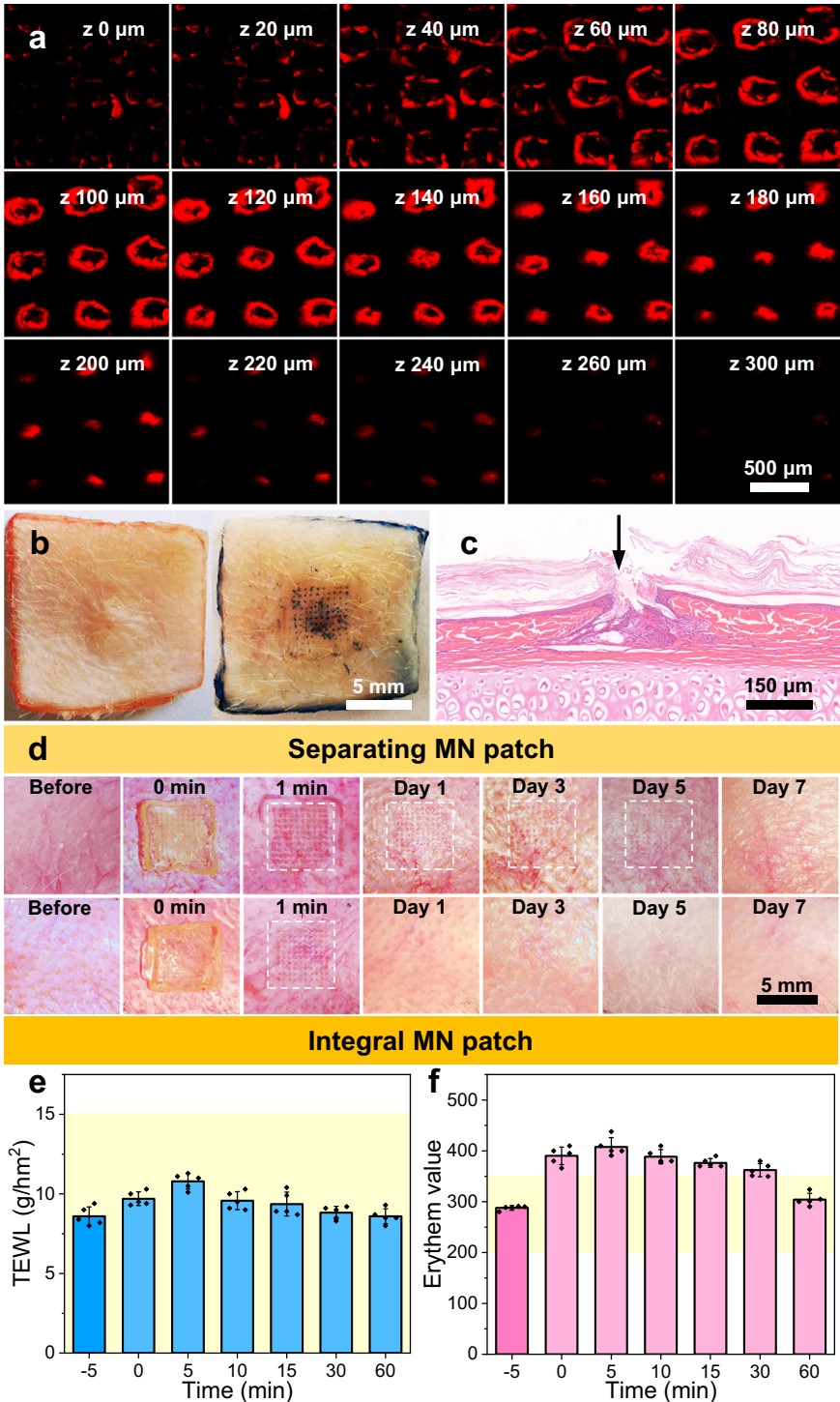

**Fig. 4 | Penetration capacity, subcutaneous retention and acute skin irritation test in humans of MN patches. a** CLSM images of porcine skin treated with RhB-loaded MN patches at varied depths after MNs insertion for 5 min ($n = 3$ independent experiments). **b** Trypan blue staining of rabbit ear HS lesions. (Left: HS lesion without treatment; Right: HS lesion treated with MNs.) **c** Image of H&E-stained rabbit ear HS tissue after MN patch insertion. **d** Photographs of rabbit ear skin tissues treated with separating MN patches (top) and integral MN patches (bottom) ($n = 3$ independent experiments). **e** TEWL and (**f**) Erythem values from the MN-treated skins of five healthy volunteers before and after insertion of separating MN patches. The yellow ranges represent the normal range. The scale bar in the last images can be applied to the others in the same panel. Images in (**b**, **c**) are representative of $n = 3$ biologically independent samples. The data in (**e**, **f**) are presented as mean ± SD ($n = 5$ independent experiments).

## RNA sequencing analysis of the drug-loaded MNs on HS treatment

To elucidate the RNA changes associated with different treatments on HS, RNA-seq analysis was performed from full thickness rabbit ear HS

model. The samples were collected from the following four groups ($n = 3$): NS, HS, drug-loaded MNs and 5-Fu injection. The results demonstrated the consistency of the global gene distribution of all samples (Fig. 7a and Supplementary Fig. 18). Clearly, the four groups

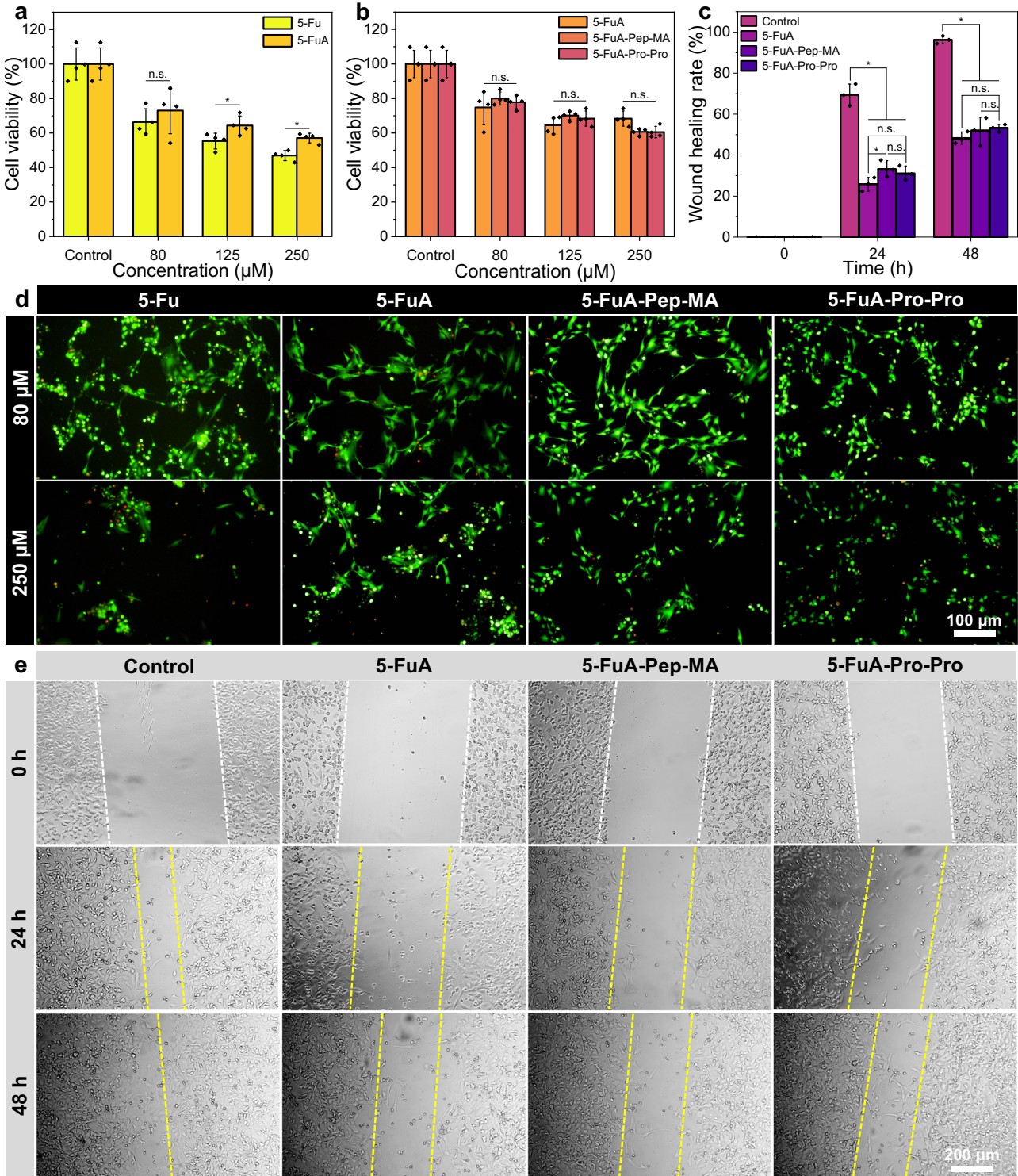

**Fig. 5 | Evaluation of fibroblast cytotoxicity and inhibition of migration activity. a** Viability of NIH/3T3 fibroblasts treated with 5-Fu and 5-FuA at different concentrations for 48 h, respectively. **b** Viability of NIH/3T3 fibroblasts after incubation with 5-FuA, 5-FuA-Pep-MA and 5-FuA-Pro-Pro solutions for 48 h, respectively. **c** Quantification of wound healing rate in scratch assay. **d** Live/dead staining fluorescence images of NIH/3T3 fibroblasts after co-incubation with 5-FuA, 5-FuA-Pep-MA and 5-FuA-Pro-Pro solutions for 48 h, respectively. **e** Photographs of wound

scratch assay at 24 and 48 h under different treatments. The scale bar in the last image can be applied to the others in the same panel. Representative images from $n = 3$ independent experiments. The data are presented as mean ± SD ($n = 4$ independent experiments in (**a**, **b**), $n = 3$ independent experiments in (**c**)). The data in **a** were analyzed by $t$-test, and the data in (**b**, **c**) were analyzed by one-way ANOVA. $P$-value: n.s. means no significance, *$P < 0.05$.

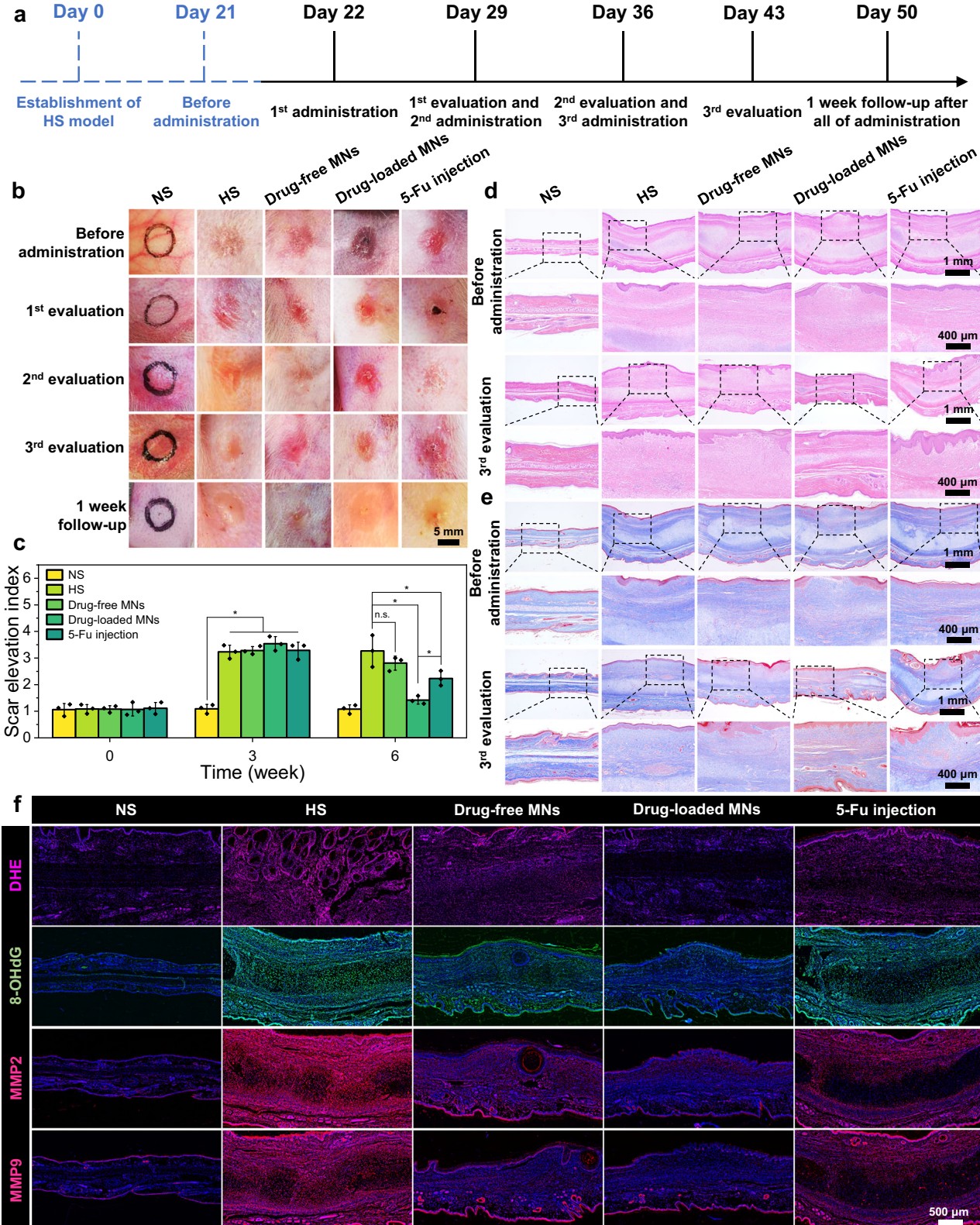

**Fig. 6 | Evaluation of in vivo therapeutic effect on rabbit ear HS under different interventions and reconstruction of pathological microenvironment.**
**a** Diagram of the establishment of HS model and evaluation of the therapeutic effect. **b** Representative photographs of HS before and after different treatments. **c** SEI of different groups after the 3rd evaluation. **d** H&E staining of HS with different treatments before and after the 3rd evaluation. **e** Masson's trichrome staining of HS with different treatments before and after the 3rd evaluation. **f** Fluorescence and immunofluorescence staining for DHE, 8-OHdG, MMP2 and MMP9 in NS and HS tissues with different treatments. The scale bar in the last images can be applied to the others in the same panel. Images in (**b**) and (**d**–**f**) are representative of $n = 3$ biologically independent samples. The data in (**c**) are presented as mean ± SD ($n = 3$ biologically independent samples). The data in (**c**) were analyzed using one-way ANOVA. $P$-value: n.s. means no significance, $*P < 0.05$.

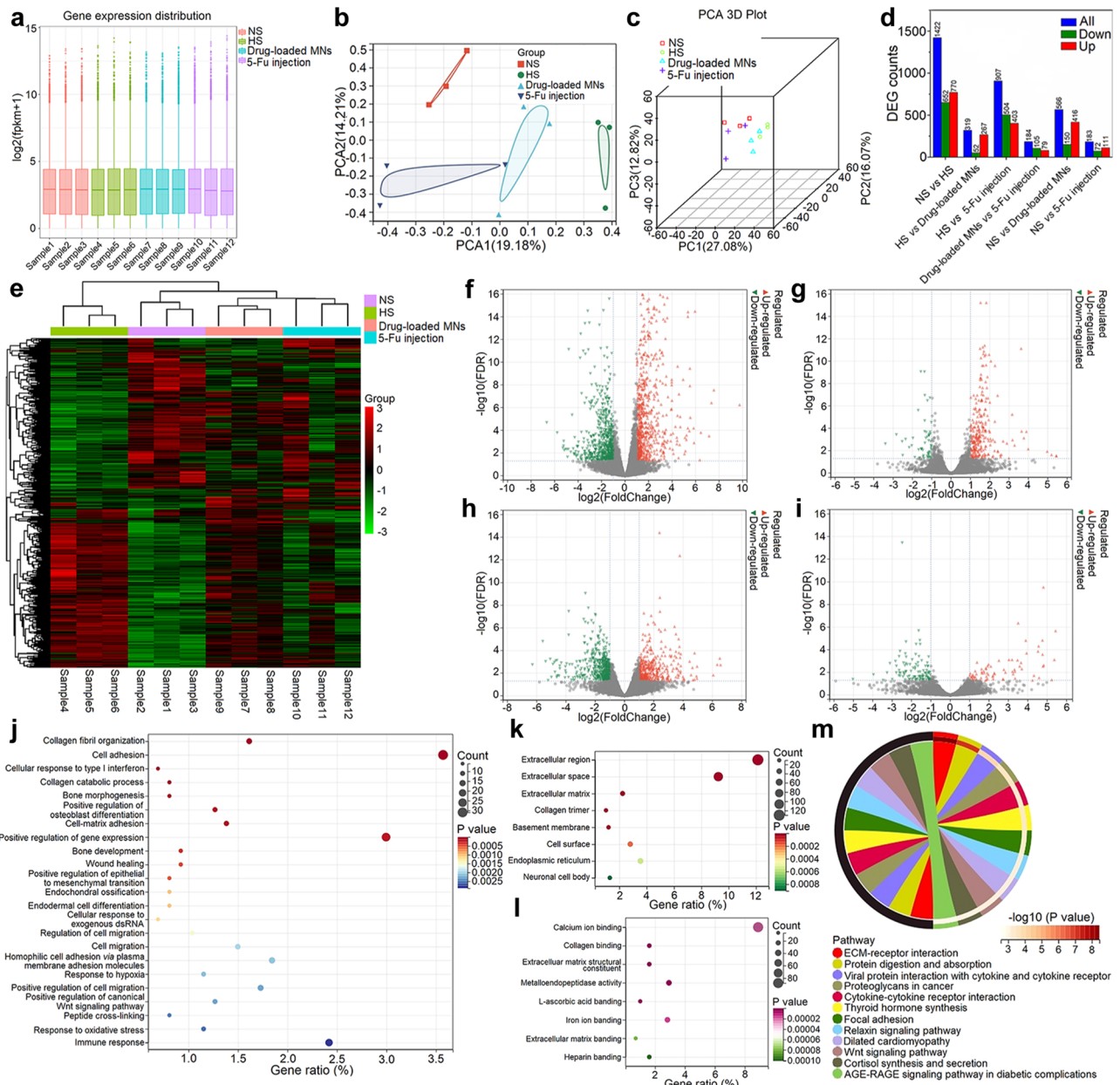

**Fig. 7 | The gene distribution and functional annotation analysis in full thickness rabbit ear HS model. a** Global gene expression distribution across 12 samples (4 groups, *n* = 3 biologically independent samples). The lower bound of box represents the first quartile (Q1; the 25th percentile) of the data, the line in the middle of the box represents the median (Q2; the 50th percentile) of the data, the higher side represents the third quartile (Q3; the 75th percentile) of the data, the lower whisker shows the minima and its variability, and the higher whisker shows maxima and its variability. **b** 2D PCA plot of the relationships among all samples. **c** 3D PCA plot indicating the separation of all samples into 4 clusters. **d** Barplot of DEGs between different groups. **e** Heatmap of DEGs among NS, HS, drug-loaded MNs and 5-Fu injection groups. **f** Volcano plot for DEGs between HS and NS groups. The green plots represent down-regulated genes, and red plots indicate up-regulated genes. **g** Volcano plot between drug-loaded MNs and HS groups. **h** Volcano plot between 5-Fu injection and HS groups. **i** Volcano plot between drug-loaded MNs and 5-Fu injection groups. **j** The significantly enriched BP of GO terms between HS and NS groups. **k** The significantly enriched CC of GO terms between HS and NS groups. **l** The significantly enriched MF of GO terms between HS and NS groups. **m** The most significantly enriched KEGG signaling pathways between HS and NS groups. The data in (**j**–**m**) were analyzed by hypergeometric distribution. The *P*-values were adjusted using the Benjamini-Hochberg method.

were separated in 2D principal component analysis (PCA) diagram as well as 3D PCA plot (Fig. 7b, c). NS and HS groups were located on opposite sides of the PCA map, while drug-loaded MNs and 5-Fu injection groups fell right in the middle. These results indicated the four groups had distinct gene expression pattern. After drug-loaded MNs and 5-Fu injection treatments, the gene profiles of HS could change towards the gene expression of NS.

The differentially expressed genes (DEGs) were subsequently identified among groups. The DEGs between HS and NS groups were

obtained from the gene profiles, revealing the up-regulated expression of 770 genes and down-regulated expression of 652 genes (Fig. 7d and f and Supplementary Fig. 19). Then, the DEGs between drug-loaded MNs/5-Fu injection and HS groups were identified. Specifically, 319 DEGs between drug-loaded MNs and HS groups included 267 genes up-regulated and 52 genes down-regulated (Fig. 7d and g and Supplementary Fig. 20); 907 DEGs between 5-Fu injection and HS group included 403 genes up-regulated and 504 genes down-regulated (Fig. 7d and h and Supplementary Fig. 21).

In addition, a total of 184 commonly DEGs between drug-loaded MNs and 5-Fu injection groups were detected, including 79 genes up-regulated and 105 genes down-regulated (Fig. 7d and i and Supplementary Fig. 22). The heatmap displayed the DEGs among all groups (Fig. 7e).

Subsequently, we conducted the functional annotation of DEGs among groups. Firstly, we investigated the pathological changes between rabbit HS and NS groups. gene ontology (GO) and Kyoto encyclopedia of genes and genomes (KEGG) pathway analysis were carried out with the identified 1422 DEGs between HS and NS (Fig. 7m). The GO analysis consisted of biological process (BP, Fig. 7j), cellular component (CC, Fig. 7k) and molecular function (MF, Fig. 7l). These enriched pathways were consistent with the HS condition in human, including collagen organization, bone morphogenesis, hypoxia, Wnt signaling pathway and immune response. Therefore, the HS models in rabbit ear could faithfully represent the pathophysiological process of HS in humans.

Then, we explored the underlying molecular changes of HS tissues under the drug-loaded MNs and 5-Fu injection treatment. The Venn diagram software was used to identify the commonly DEGs between drug-loaded MNs and 5-Fu injection modalities, and a total of 102 DEGs were commonly shared (Fig. 8a). To further detect the differences in therapeutic mechanism between drug-loaded MNs and 5-Fu injection on HS, gene set enrichment analysis (GSEA) analyses of GO and BioCarta genesets was used for functional enrichment analyses at an overall level. Notably, the down-regulated GO terms enrichment between drug-loaded MNs and 5-Fu injection were identical, including collagen fibril organization and ECM structural constituent (Fig. 8b, c). However, the up-regulated GO terms were significantly different between the two modalities. The enrichment of most up-regulated GO terms in drug-loaded MNs group was related to the keratinization, keratinocyte differentiation and epidermis development, while those in 5-Fu injection group were mainly enriched in structural constituent of ribosome, ribosome subunit and ATP synthesis coupled electron transport. As shown in Fig. 8d, the most enriched BioCarta pathways between drug-loaded MNs and HS groups included down-regulated BAD pathway, nuclear factor of activated T cells (NFAT) pathway and IGF1R pathway, and up-regulated Toll pathway, IL1R pathway and keratinocyte pathway. In comparison, the most enriched BioCarta pathways between 5-Fu injection and HS groups included down-regulated ALK pathway, TGF-β pathway and NFAT pathway, and up-regulated TID pathway, INFLAM pathway and NKT pathway (Fig. 8e). These results indicated the different mechanism between drug-loaded MNs and 5-Fu injection on HS treatment.

The PPI Network was constructed by STRING and visualized by Cytoscape. The MCODE plugin was used to identify gene cluster modules. The top module was shown in Fig. 8f, which weighed 6.727 score and comprised 12 nodes and 37 edges. Meanwhile, we identified the top 20 hub genes from six algorithms of cytoHubba including MCC, MNC, Degree, EPC, Radiality, and Stress algorithms (Supplementary Table 1). Among them, 12 hub genes were identified by intersecting the results (Fig. 8g). These genes are the most important genes in PPI network, including *FLG2*, *DSG1*, *EVPL*, *KRT10*, *CDSN*, *DSC1*, *SPINK5*, *PKP1*, *ABCA12*, *JUP*, *PNPLA1* and *NIPAL4*, which are associated with keratinocyte constituent and differentiation. Therefore, keratinocyte may play an important role in the therapeutic mechanism of drug-loaded MNs.

### Single-cell RNA sequencing analysis of the drug-loaded MNs on HS treatment

Bulk RNA-seq can reflect the average gene expression level in cell populations, which can be used to explore differences in gene expression between different treatments. The results of bulk RNA-seq suggested that drug-loaded MNs could reverse skin fibrosis through regulating keratinocyte differentiation. Therefore, it is also essential to investigate the mechanisms of drug-loaded MNs in HS tissue at a single cell level, especially the interactions between fibroblasts and keratinocytes. Rabbit ear scar tissues with no treatment (HS group) and drug-loaded MNs treatment (drug-loaded MNs group) were collected for scRNA-seq (Fig. 9a). Then, we excluded some cells and limited the percentage of mitochondrial genes to ensure the reliability of samples (Supplementary Fig. 23). A total of 29,634 cells (HS: 14,481; drug-loaded MNs: 15,153) were obtained (Fig. 9b). All the cells were clustered by uniform manifold approximation and projection (UMAP) clustering and divided into 20 cell clusters (Fig. 9c), which were annotated into 9 major cell types (Fig. 9d) by classic markers and particular transcriptional signatures. A heat map displayed the correlations between cell clusters and annotated cell types (Supplementary Fig. 24).

Keratinocytes and fibroblasts are the major cell types of epidermis and dermis, respectively. To reveal the functions of epidermis and dermis contributing to the drug-loaded MNs treatment, DEGs in keratinocytes and fibroblasts between HS and drug-loaded MNs groups were explored. Specifically, the significantly expressed DEGs in keratinocytes were mainly enriched in structural molecule activity, ribosome, structural constituent of ribosome, fibrillar collagen trimer, banded collagen fibril, etc., revealing that keratinocytes could regulate the function of fibroblasts (Fig. 9e). The significantly expressed DEGs in fibroblasts were associated with cornified envelope, structural molecule activity, keratinocyte differentiation, epithelium development, epithelial cell differentiation, etc., suggesting that fibroblasts could regulate the development and differentiation of keratinocytes (Fig. 9f). In addition, the top 50 DEGs in keratinocytes and fibroblasts between drug-loaded MNs and HS groups were displayed in heatmaps, respectively (Supplementary Figs. 25 and 26). Therefore, keratinocytes and fibroblasts could regulate the behavior of each other, implying that mutual interactions between keratinocytes and fibroblasts could participate in the drug-loaded MNs treatment.

Subsequently, the cell-cell communication landscape between each cell type was explored. To better understand the cell-cell communication, CellChat tool was used to quantitatively infer the intercellular communication based on scRNA-seq data. We first inferred the number and weights/strength of interactions among different cell population. Most interactions were observed among fibroblasts, keratinocytes, endothelial cells, macrophages, mast cells, and peri-islet Schwann cells in drug-loaded MNs group (Fig. 9g, Supplementary Fig. 27). Considering the weights/strength of ligand-receptors, fibroblasts played a pivotal role in the HS tissues with drug-loaded MNs treatment. More importantly, the strength of intercellular interactions between fibroblasts and keratinocytes was more abundant than those among other cell types (Fig. 9h). Then, heat maps displayed the number and strength of interactions between different cell types, indicating that fibroblasts and keratinocytes emerged as one of the major sources and targets in drug-loaded MNs group (Supplementary Fig. 28). These results suggested that the interactions between fibroblasts and keratinocytes played a central role in drug-loaded MNs treatment.

We further explored the ligand-receptor pairs involving in the major cellular crosstalk. The results showed that fibroblasts, keratinocytes, and endothelial cells were more active in HS tissue with drug-loaded MNs treatment. Communication probabilities mediated by ligand-receptor pairs between different cell types were compared (Fig. 9i). It was found that fibroblasts interacted strongly with keratinocytes in drug-loaded MNs group. Notably, strong communication probabilities mostly occurred between ligands (collagens, *COL6A2*, *COL1A2*, *COL1A1*) and receptors (syndecan 1 *SDC1*, cluster of differentiation *CD44*). These ligands and receptors dominated the active signaling between fibroblasts and keratinocytes. As detailed in bubble plots, a denser interaction network between fibroblasts and keratinocytes in heparan sulfate proteoglycans 2 (HSPG2)-dystroglycan 1(DAG1) signaling pathway could be observed (Fig. 9j). The signal was

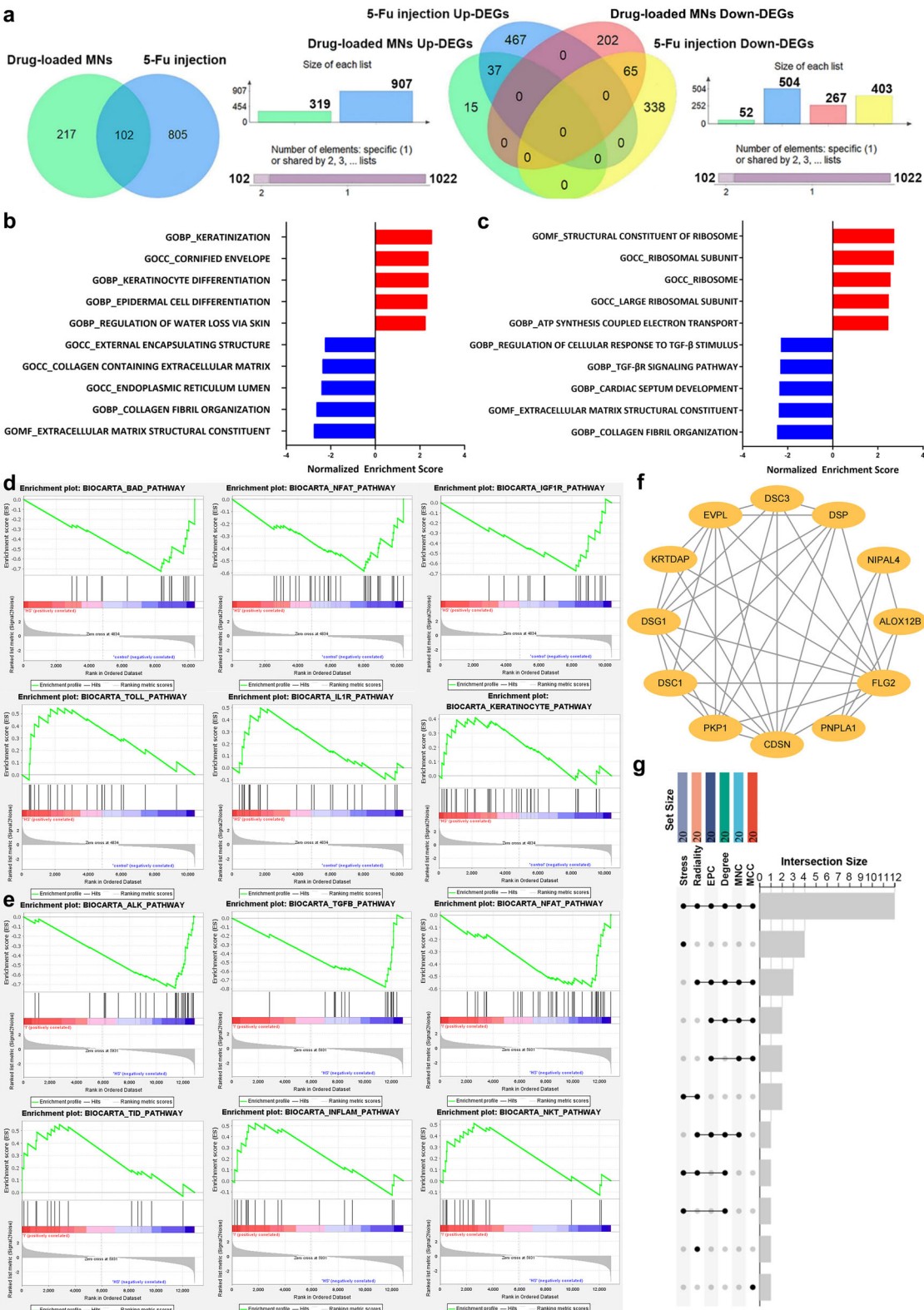

**Fig. 8 | The function annotation of DEGs in full thickness rabbit ear HS model with drug-loaded MNs and 5-Fu injection treatment. a** Venn diagram showed the intersection of DEGs obtained by drug-loaded MNs compared with HS and 5-Fu injection compared with HS analysis. **b** GSEA results for up-regulated and down-regulated GO terms compared between drug-loaded MNs and HS groups. **c** GSEA results for up-regulated and down-regulated GO terms compared between 5-Fu injection and HS groups. **d** GSEA enrichment plot showing the top ranked down-regulated and up-regulated BioCarta pathways between drug-loaded MNs and HS groups. **e** GSEA enrichment plot showing the top ranked down-regulated and up-regulated BioCarta pathways between 5-Fu injection and HS groups. **f** The sub-network of the 12 significant genes with higher degrees in the PPI network selected by MCODE. **g** Vertical bars of upper plot showing the intersection of the 20 hub genes screened by six algorithms in cytoHubba.

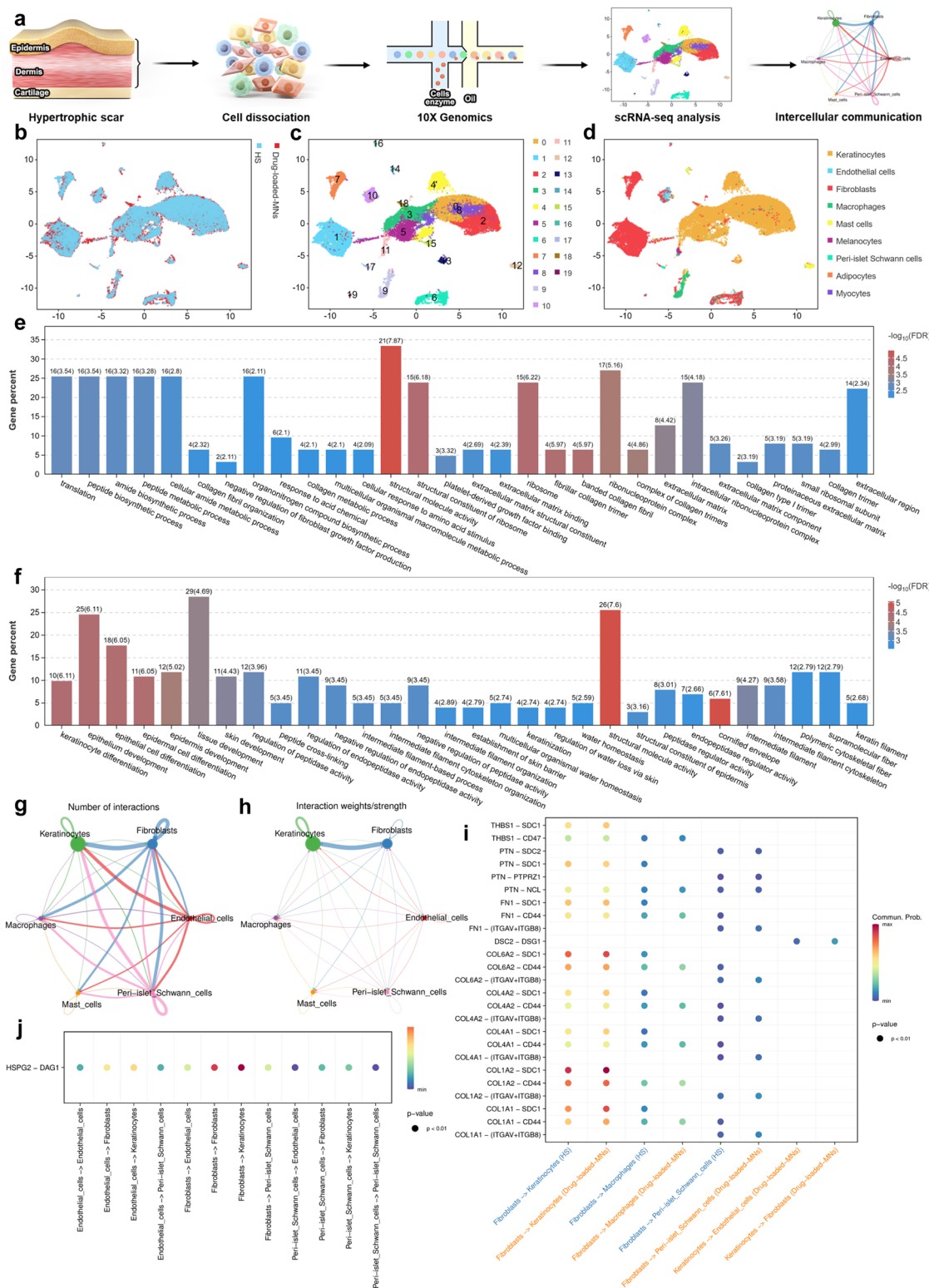

**Fig. 9 | Single-cell RNA-seq of epidermis and dermis in rabbit ear HS model with no treatment and drug-loaded MNs treatment. a** Schematic representation of scRNA-seq procedure in epidermis and dermis of rabbit ear HS tissue. **b** UMAP plot of the profiled single cells. Cells with the same colors are from the same group. **c** UMAP clustering of all the cells by classic markers and particular transcriptional signatures. **d** UMAP plot of all the cells annotated into 9 major cell types. **e** Bar plots of significantly enriched GO terms of the DEGs in keratinocytes. **f** Bar plots of significantly enriched GO terms of the DEGs in fibroblasts. **g** Cell-cell

communications between any two identified cell types illustrated with the interaction numbers for each of the cell types in drug-loaded MNs group. **h** Cell-cell communications between any two identified cell types illustrated with the interaction strength for each of the cell types in drug-loaded MNs group. **i** The significantly ligand-receptor pairs of mostly communicated cell types in HS and drug-loaded MNs groups. **j** Bubble plot of the expression level of HSPG2-DAG1 pair among different cell types in drug-loaded MNs group. The *P*-values in (**i, j**) were computed from one-sided permutation test.

further amplified when we identified dysfunctional signaling between drug-loaded MNs and HS groups (Supplementary Fig. 29). Therefore, the interactions between fibroblasts and keratinocytes via HSPG1-DAG2 were the molecular underpinnings during the drug-loaded MNs treatment.

## Discussion

At present, dissolving MNs as transdermal drug delivery platforms exhibit instant or burst drug release, producing high dose in the local lesions, confronting a potential biosafety risk[45,46,49]. Moreover, HS treatment requires slow and sustained drug release to enhance the therapeutic efficacy through long-term administration[11,15]. Although hydrophobic and biodegradable polymers have been used as MN matrix for sustainable drug release, the short half-life and low bioavailability of 5-Fu still restrict the clinical application of 5-Fu physically loaded MNs[11]. To address these problems, we have developed a separating MN drug delivery system consisting of GelMA and 5-FuA prodrug in response to endogenous stimuli (MMP2, MMP9 and ROS) to remodel the pathological microenvironment for HS treatment.

After penetration in the HS lesions, the base layer of the separating MN patch prepared by the double-step perfusion method was easy to peel off, leaving the tips retained in the HS tissues. The photopolymerization and crosslinking of 5-FuA prodrug in GelMA-based MN matrix endowed MN tips with controlled and slow drug release in response to endogenous stimulation of pathological microenvironment. Compared with the traditional intralesional injection, the drug dosage was reduced and there was no secondary damage during treatment. Notably, the drug-loaded MNs in response to endogenous stimuli (MMP2, MMP9 and ROS) could decrease the ROS level and consume MMP2 and MMP9, leading to remodel the pathological microenvironment of HS tissues.

Drug-loaded MNs and 5-Fu injection showed some difference in underlying mechanisms for HS treatment demonstrated by bulk RNA-seq analysis. Traditional 5-Fu injection could inhibit fibroblast proliferation and collagen deposition via down-regulated TGF-β pathway and collagen fibril organization process, and induce inflammatory response via up-regulated TID, INFLAM and NKT pathway in HS treatment. In comparison, drug-loaded MNs could significantly promote fibroblasts apoptosis and decrease collagen fiber deposition through down-regulated BAD, IGF1R pathways and collagen fibril organization process, simultaneously stimulating the inflammatory response and mediating keratinocyte differentiation through up-regulated TOLL, IL1R and keratinocyte pathways. BAD pathway is not reported in the pathogenesis of HS, but it has been involved in the fibrotic process, such as lung and liver fibrosis[26,50,51]. The results of these studies support specific BCL-2 family pro-survival factors could promote myofibroblast survival to induce tissue fibrosis. IGF-1 and IGF-2 are potent mitogens and inhibitors of apoptosis. Activation of the IGF1R by binding of IGF-1 and IGF-2 plays important roles in induction of proliferation, inhibition of cell apoptosis and induction of differentiation[52]. In addition, IGF1R is highly expressed in dermal fibroblasts of HS and keloid[53], and IGF-1-induced AKT/mTOR/HIF-1α signaling pathway is involved in the pathogenesis of keloid[54]. Therefore, down-regulation of BAD and IGF1R pathways could promote fibroblast apoptosis, inhibit collagen synthesis and reverse skin fibrosis. Although both drug-loaded MNs and 5-Fu injection could inhibit collagen fibril organization and ECM production, drug-loaded MNs could mediate keratinocyte differentiation and regulate transepidermal water loss. It has been reported that keratinocytes are involved in the development of pathological fibrosis in HS tissues by regulating the behavior of dermal fibroblasts and myofibroblasts[55]. Disruption of the skin barrier could result in transepidermal water loss, increase the expression and secretion of HMGB1 in keratinocytes, subsequently activating dermal fibroblasts and leading to fibrosis[56,57]. Thus, we speculated that drug-loaded MNs could decrease collagen deposition by mediating keratinocyte differentiation and epidermis development, verified by the up-regulated expression of keratinocyte pathway when drug-loaded MNs were introduced.

ScRNA-seq is a novel technique for sequencing the transcriptome at the single-cell level, which can help us understand pathogenesis and develop more precise therapeutic strategies. Due to the results of bulk RNA-seq, we further investigate the interactions between fibroblasts and keratinocytes of drug-loaded MNs treatment in HS tissue at a single cell level. ScRNA-seq data analysis on rabbit ear HS tissue with no treatment and drug-loaded MNs treatment was performed. HSPG2 is a ubiquitous heparan sulfate proteoglycan, which is involved in tissue regeneration and wound healing[58,59]. It has been found at the cell surface and in the ECM, they interact with a plethora of ligands. HSPG2 could regulate fibrogenesis through growth factor binding and signaling, ECM interactions and activities, and apoptosis inhibition of fibroblasts. In our study, HSPG2-DAG1 was found to be significantly expressed signaling pathways at the level of ligand-receptor communication between fibroblasts and keratinocytes. Thus, we inferred that fibroblasts could interact with keratinocytes via HSPG2-DAG1 signaling pathway, which contributed to inhibit HS tissue with drug-loaded MNs treatment.

In short, we hypothesize that drug-loaded MNs could inhibit fibroblast hyperplasia, stimulate the inflammatory response and mediate keratinocyte differentiation to reduce collagen fiber deposition. In addition, drug-loaded MNs could remodel the pathological microenvironment of HS tissues by ROS scavenging and consumption of MMP2 and MMP9. Therefore, the well-designed MN patches as a minimally invasive and painless manner provides convenient, safe and efficient HS therapy in clinical application, leading to good patient compliance for long-term HS treatment. More importantly, this study demonstrates that fibroblasts are the major cells expressing ligands and receptors that are actively interacted with keratinocytes in HS tissue, which may help us better understand the therapeutic mechanism of drug-loaded MNs and provide more efficient strategies for HS treatment.

## Methods
### Ethical statement
The human study protocol was approved by the Institutional Review Board at Wuhan Union Hospital, Tongji Medical College, Huazhong University of Science and Technology (HUST) (Approval No. [2022] IACUC Number: 0509), and were carried out in accordance with the Declaration of Helsinki. Meanwhile, the volunteers were allowed to participate in the experiment after providing informed consent.

All the animal experiments were performed in line with the guidelines of "The Care and Use of Laboratory Animals in HUST", as well as approved by the Institutional Animal Care and Use Committee of Tongji Medical College, HUST (Approval No. [2021] IACUC Number: 2751), and were performed in accordance with the National Institutes of Health guidelines for the care and use of laboratory animals.

### Cell lines and animals
Standard fibroblast cell line (NIH/3T3) was supplied by the Cell Bank of the Chinese Academy of Sciences (Shanghai, China). The human hypertrophic scar fibroblast (HSFb) was obtained from Hubei Engineering Research Center for Skin Repair and Theranostics (Wuhan, China). Cells were cultured in DMEM (containing 10% FBS and 1% penicillin-streptomycin solution) at 37 °C in a humid atmosphere with 5% $CO_2$.

Fifteen female rabbits (*Oryctolagus cuniculus*, 10-12 week, ~2.3-2.7 kg) were purchased from Experimental Animal Center of Tongji Medical College, HUST (Wuhan, China). All experimental animals were caged individually, fed at the temperature of $25 \pm 2$ °C and relative

humidity of 55% ± 2%, and maintained under standard conditions with a 12-h light/dark cycle. Ten female BALB/c mice (8-10 week, -18-20 g) were purchased from Liaoning Changsheng Biotechnology Co, td. (Benxi, China). The mice were kept at the temperature of $25 \pm 2 °C$ and relative humidity of 55% ± 2% under standard conditions. All the mice were fed ad libitum, and allowed free access to water.

## Synthesis and characterization of 5-Fuorouracil-1-acetic acid (5-FuA)

5-Fu (13.0 g, 1.0 equiv.) was dissolved in KOH (22.4 g, 4.0 equiv.) aqueous solution (70 mL) at 60 °C. Then, 30 mL of bromoacetic acid (20.84 g, 1.5 equiv.) aqueous solution was slowly added dropwise. The mixed solution was stirred at 60 °C for 8 h, and cooled to room temperature. The pH value was initially adjusted to -5.5 using dilute HCl solution, and the obtained solution was placed at 4 °C for 2 h, accompanied with precipitation. Then, the pH value of the supernatant was adjusted to -1 using the concentrated HCl, and the obtained solution was placed at 4 °C overnight. The white precipitate was collected by filtration and vacuum drying. After being washed for three times with distilled water, the precipitate was recrystallized in water. Needle-like crystals were collected by filtration and vacuum drying, which were stored at 4 °C before use. 5-FuA was characterized by $^1$H NMR and electrospray time-of-flight high-resolution mass spectrometry (ESI-HRMS). 5-FuA was dissolved in $D_2O$. $^1$H NMR spectrum was obtained on a dual-channel fully digitized Fourier superconducting nuclear magnetic resonance spectrometer (AV400, BRUKER, Switzerland). ESI-HRMS data were obtained on an electrospray time-of-flight high-resolution mass spectrometer (micrOTOF II, BRUKER, USA). $^1$H NMR (400 MHz, $D_2O$): δ 7.82 (d, $J = 5.8$ Hz, 1H), 4.54 (s, 2H). HRMS (ESI) m/z: [M-H]$^-$ calcd for $C_6H_4FN_2O_4$, 187.0161; found, 187.0197.

## Synthesis and characterization of 5-FuA-Pep-MA

5-FuA-Pep-MA was synthesized on Rink amide-MBHA resin (0.66 mmol/g, 1.0 equiv.) using a standard Fmoc solid-phase peptide synthesis (SPPS) procedure. Briefly, the typical SPPS procedure included deprotection of the N-terminal Fmoc group with 20 % 4-methyl-piperidine in DMF (2 × 20 min) and amide couplings (2 × 4 h) with the mixture of Fmoc-protected amino acid (Fmoc-Lys(Alloc)-OH, Fmoc-Pro-OH, Fmoc-Pro-OH, Fmoc-Pro-OH, respectively, 3.0 equiv.), PyBOP (3.0 equiv.) and DIPEA (6.0 equiv.). 5-FuA was then coupled to the N-terminal of the tetrapeptide (PPPK) for 8 h with the mixture of 5-FuA (4.0 equiv.), PyBOP (4.0 equiv.) and DIPEA (8.0 equiv.). After 5-FuA coupling, the deprotection of Alloc group was achieved with the addition of Pd(Ph$_3$P)$_4$ (0.1 equiv.) and PhSiH$_3$ (24 equiv.) (2×10 min), followed by amide coupling (2×4 h) with the mixture of methylacrylic acid (3 equiv.), PyBOP (3.0 equiv.) and DIPEA (6.0 equiv.). 5-FuA-Pep-MA was cleaved from the resin (1×3 h) with the addition of TFA/TIS/ $H_2O$ (95%: 2.5%: 2.5%) mixture, which was condensed and precipitated in ice ether. The crude product was then purified by the medium pressure purification chromatography system (MPLC; BLUE-OCTO-PUS, Agela Technologies, China) on C18-reversed phase silica gel. The eluting solvent system consisted of (A) 0.05% TFA in water and (B) 0.05% TFA in methanol. After acidification with dilute hydrochloric acid, 5-FuA-Pep-MA was obtained through freeze-drying and stored at −20 °C before use. 5-FuA-Pep-MA was characterized by $^1$H NMR and ESI-HRMS. 5-FuA-Pep-MA was dissolved in $D_2O$. $^1$H NMR spectrum was obtained on a dual-channel fully digitized Fourier superconducting nuclear magnetic resonance spectrometer (AV400, BRUKER, Switzerland). ESI-HRMS data were obtained on an electrospray time-of-flight high-resolution mass spectrometer (micrOTOF II, BRUKER, USA). $^1$H NMR (400 MHz, $D_2O$): δ 7.76 (d, $J = 5.8$ Hz, 1H), 5.41 (d, $J = 5.3$ Hz, 1H), 4.73 (d, $J = 5.3$ Hz, 1H), 4.70 (m, 1H), 4.59 (m, 1H), 4.41 (dd, $J = 8.5, 5.4$ Hz, 1H), 4.21 (dd, $J = 8.7, 5.5$ Hz, 1H), 3.80 (m, 2H), 3.66 (m, 3H), 3.64 (m, 3H), 3.25 (m, 1H), 3.18 (t, $J = 6.9$ Hz, 1H), 2.29 (m, 1H), 2.03 (m, 8H), 1.90 (m, 3H), 1.76 (m, 3H), 1.52 (dq, $J = 15.8, 7.3$ Hz, 2H), 1.38 (m, 2H), 1.10 (m, 2H).

HRMS (ESI) m/z: [M+Na]$^+$ calcd for $C_{31}H_{43}FN_8NaO_8$, 697.3107; found, 697.3080.

## Fabrication and morphology of MN patches

0.4 g of GelMA and 10 mg of photoinitiator (Irgacure 2959) were dissolved in 1.5 mL of PBS at 50 °C, followed by the addition of 5-FuA-Pep-MA PBS solution (0.5 mL) at different concentrations under vigorous shaking. The MN mould was subsequently immersed into above precursor solution upon sonication at 50 °C for 0.5 h to facilitate the filling of the precursor solution into the mould. The excess solution was then removed, only leaving the precursor solution in the cavity. The MN mould was exposed to UV light (365 nm, 400 mW/cm²) for different time.

100 μL of gelatin solution (0.2 g/mL) in PBS was added to construct the base layer, which could completely cover the MN tips. The bubbles were removed by centrifugation. After drying at room temperature for 24 h in the dark, MNs patches were stripped by demoulding and stored at 4 °C before use. Morphology and the size of MN tips were characterized with a digital microscope (AM4113ZTL Dino-Lite Premier, AnMo Electronics Corporation, China). Meanwhile, FITC-labeled gelatin was used to prepare the base layer of MN patch, which was subjected to fluorescence microscope (IX71, Olympus, Japan).

## Separable capacity of MN patches

The MN patch was pressed into an agarose gel (3 wt.%). 5 min later, the MN base layer was carefully removed from the surface of the agarose gel, which was then subjected to a digital microscope (AM4113ZTL Dino-Lite Premier, AnMo Electronics Corporation, China). At the same time, integral MNs underwent similar operation as control.

## In vivo human skin irritation study

An in vivo human skin irritation study was performed in 5 healthy human volunteers: a 23-year-old woman, a 25-year-old woman, a 25-year-old man, a 27-year-old man, and a 30-year-old man. The in vivo skin irritation study was performed to further explore the skin irritation of the separating gelatin MNs used in human, which included transepidermal water loss (TEWL), skin erythema (erythem value), skin melanin (melanin value) and corneum water (corneometer value) measurement. The separating MNs were inserted into the skin of volunteers for 5 min. The CUTOMETER DUAL MPA 580 (Courage + Khazaka electronic GmbH, Cologne, Germany) was used to determine TEWL, erythem value, melanin value, and corneometer value of skins before (prior to MN insertion) and after MN treatment. After removing the base layer of the separating MNs, measurements were performed on the treated skin at different time intervals, i.e., 0 min, 1 min, 3 min, 5 min, 10 min, 15 min, 30 min, and 60 min, respectively.

## Establishment of HS model in rabbits

We investigated therapeutic effects of separating MNs on HS model in rabbit ears. Among the 15 rabbits, three rabbits were used as the control group (normal skin, NS), and the other 12 rabbits were used to establish HS model. The rabbit ear HS model was established according to a standard method, i.e., 4 HS lesions on each ear and 96 in total. Briefly, the rabbits were anesthetized with sodium pentobarbital (30 mg/kg) and operated under sterile conditions. Then, 4 round-shaped wounds of 7 mm diameter were created on the ventral surface of each ear, which were kept away from the central auricular artery and the periauricular vein. The epidermis, dermis, and perichondrium in each wound was removed thoroughly. After the procedures, the wounds were washed with 0.1% benzalkonium chloride solution to sterilize. In the following 2 weeks, after the rabbits were anesthetized, the newly formed wound tissues were removed to accelerate the formation of HS. Twenty-one days after wound resection, the thickness of HS lesions was -3 times above NS, and the HS models were successfully established.

## In vivo HS inhibition study

The three rabbits without treatment were taken as the control group (group 1, NS). The other twelve rabbits with HS lesions were randomly divided into 4 groups (three rabbits in each group): group 2 (HS), group 3 (drug-free MNs), group 4 (drug-loaded MNs), and group 5 (5-Fu injection). Administration started from the 22nd day after skin resection, once a week for 3 consecutive weeks (on the 22nd, 29th and 36th day). For group 1 and 2, no treatment was given. For group 3, drug-free MNs was used to evaluate the impact of MNs on HS treatment. For group 4, one patch of drug-loaded MNs was used in each HS lesion. In group 5, 0.1 mL of 5-Fu solution (120 μg) was injected in each HS lesion. The macroscopic images were recorded on the 21st day (before administration), the 29th day (1 week after the 1st administration), the 36th day (1 week after the 2nd administration), the 43rd day (1 week after the 3rd administration) and the 50th day (1 week follow-up).

## Histological analysis

On the 21st, 29th, 36th and 43rd day, three HS tissue sections were collected randomly in each group for histological analysis. The tissue sections were fixed in 4% paraformaldehyde and embedded in paraffin wax. Then, H&E and Masson's trichrome staining were performed to investigate the collagen deposition and therapeutic effects. The histological slides were photographed by a microscope (BX51, Olympus, Japan). Image J software was used to calculate the scar elevation index (SEI) as follows: $SEI = H/H_0$. Where, $H$ represents the height between the highest point to the surface of cartilage in HS, and $H_0$ represents the height from the stratum corneum to the surface of the cartilage at the bottom of HS. Determination of SEI was performed under the supervision of two dermatologists from Union Hospital, Tongji Medical College, HUST, and the values were then averaged.

## Bulk RNA-seq and scRNA-seq

On the 43rd day (1 week after the 3rd administration), the tissues from different groups were harvested to perform bulk RNA-seq and scRNA-seq. RNA-seq analysis of full thickness scar tissues of rabbit ears from group 1, 2, 4 and 5 were performed. Total RNA was used as input material for the RNA sample preparations. RNA integrity was assessed using the RNA Nano 6000 Assay Kit of the Bioanalyzer 2100 system (Agilent Technologies, CA, USA). Library quality was assessed on the Agilent Bioanalyzer 2100 system. High throughput sequencing was completed with an Illumina NovaSeq 6000 System (Illumina Inc., San Diego, CA). ScRNA-seq was performed with 2 rabbit ear scar tissues without treatment (group 2) and 2 rabbit ear scar tissues with drug-loaded MNs treatment (group 4). The ventral epidermis and dermis of the rabbit ear scar tissues were saved and cut into small pieces to further disassociated into single cell suspension. Cell viability was detected by trypan blue staining. Single-cell gel beads-in-emulsion generation, barcoding, sample cleanup, cDNA amplification, and cDNA library construction were performed. Subsequently, cDNA library was sequenced and bioinformatic information was analyzed.

## Statistical analysis

All data were expressed as mean ± standard deviation (SD). Independent samples $t$-tests and Mann-Whitney $U$-tests were used to evaluate differences between two groups, and one-way analysis of variance (ANOVA) test was used to evaluate differences among multiple groups. Image J software (Java-8 versions) was used to calculate wound healing rate of in vitro wound scratch assay and scar elevation index of in vivo histological analysis. We performed statistical analysis with SPSS (version 25.0, International Business Machines Corporation, USA). Two-sided $P$-value < 0.05 was considered statistically significant.

## Reporting summary

Further information on research design is available in the Nature Portfolio Reporting Summary linked to this article.

## Data availability

The authors declare that all the data supporting the findings of this study within the paper and the Supplementary Information are available from the corresponding author upon request. The bulk RNA sequencing data have been submitted to the NCBI Gene Expression Omnibus (GEO) datasets with accession number GSE254263. The single cell RNA sequencing data have been submitted to the NCBI GEO datasets with accession number GSE254543.

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

## Acknowledgements

The authors gratefully acknowledge the financial support from the National Natural Science Foundation of China (52173124, H.J.; 82103769, N.Z.; 82130089, J.T. and 82071844, Y.L.), the Fundamental Research

Funds for the Central Universities (2172019kfyXJJS070, H.J.) and Key Research and Development Program of Hubei Province (YFXM2021000203, J.T.). The authors are grateful to HUST Analytical and Testing Center for their supports on its facilities.

## Author contributions

Z.-R.Y., H.S., H.J., J.T. and J.Z. conceived the idea and designed the experiments; Z.-R.Y. prepared and characterized the MNs; H.S. and Z.-R.Y. performed in vitro and in vivo experiments; Z.-R.Y. and H.S. conducted the experiments with assistance from J.-W.F., K.D., H.Q., T.M. and N.L.; Y.L., L.Y. and N.Z. contributed to analysis and discussion of the results; Z.-R.Y. and H.S. wrote the paper; J.T., H.J. and J.Z. revised the manuscript; J.T., H.J. and J.Z. supervised the overall study.

## Competing interests

The authors declare no competing interests.
