## [Peer Review File · Nature Communications]

Reviewers' Comments:

Reviewer #1:

Remarks to the Author:

This study developed a separating MN drug delivery system consisting of GelMA and 5-FuA prodrug in response to endogenous stimuli (MMP2, MMP9 and ROS) to remodel the pathological microenvironment for HS therapy. The novelty of this study was to develop a responsive MN system that prolonged the skin retention of 5-Fu, scavenged ROS and consumed MNPs for enhanced anti-scarring effect. However, the responsive drug release behavior was only confirmed from the in vitro experiments, and some important experiments are missing. Besides, therefore, there are still many other doubts and problems to be addressed. Therefore, I do not recommend it for acceptance by this journal. The detailed comments are shown as follows.

1. The organization of introduction should be better improved. In addition to 5-Fu, intralesional injection of triamcinolone is the first-line therapy for HS. Why 5-Fu was selected for the treatment of HS? What is the advantage of MNs for drug delivery as compared to the injection. How the BAD and IGF1R pathways participate in the pathogenesis of HS? The related information was not provided, making the design of this study not quite attractive.

2. There are some grammatical errors in the manuscript, which need to be checked again and corrected.

e.g. The crosslinking degree could be controlled by UV light irradiation to endow the MN tips with tunable mechanical and drug release properties.

e.g. To visualize the drug distribution and penetration in local skin, the fluorescence microscopy images were taken perpendicular to skin surface at different depths using confocal laser scanning microscope (CLSM).

3. There are some doubts that whether the mechanical strength of the separating MN is sufficient, since its mechanical strength is less than 50 N/patch in Fig. 3a. Besides, how the MNs achieved separation performance was not clearly provided. The advantage of separating MNs and UV-crosslinked MNs in improving the transdermal delivery efficiency was not described and related experiments were missing.

4. In Fig. 3e and 3h, the degradation rate of GelMA and the drug release rate of 5-FuA from the crosslinked GelMA/5-FuA-Pep-MA hydrogels with the UV-crosslinking duration of 45 s appeared to be approaching the plateau at 24 h, and it had better to continue studying how long it takes for the drug to achieve most release.

5. Is there a contradiction that the separating MN was able to form microholes in rabbit skin for 5 days in Fig. 4d, but trans epidermal water loss decreased rapidly within 10 minutes in Fig. 4e.

6. Fig. 4b and 4c mainly showed that the separating MN has sufficient mechanical strength to break the skin barrier of HS tissues. It seems to need more evidence in the "Penetration capacity and subcutaneous retention of separating MN patches" section.

7. In Fig. 6b, the therapeutic effect of drug-loaded MNs and 5-Fu injection was not obvious, in comparison to drug-free MNs.

8. The human hypertrophic scar fibroblast (HSFb) is usually adopted for the cellular study of HS, while NIH/3T3 fibroblasts were used to evaluate the ROS scavenging ability of 5-FuA-Pep-MA. I doubt whether this experiment is reasonable and 5-FuA-Pep-MA is available for scavenging ROS in HS tissues.

9. The reference format is inconsistent, and it had better to refer to more recent studies.

10. The resolution of Fig. 3 and Fig. 6-8 is poor.

11. The first appearance of BAD and IGF1R in the abstract and introduction should provide the full name.

(Note from the Editor: the report of Reviewer #1 with highlighted text is attached.)

Reviewer #2:

Remarks to the Author:

This study reports the use of a separating microneedle system for sustained 5-fluorouracil (5-Fu) in the hypertrophic scar (HS) treatment. 5-Fu was designed as a prodrug through binding with ROS-responsive tetrapeptide (PPPK) at the N-terminus and methacryloyl on the lysine residue, which was UV-crosslinked with GelMA as the tips of separating MN patches. Later the detached MN tips slowly released 5-Fu, which reduced abnormal fibroblast proliferation and collagen fiber deposition.

In general, the experiments were designed well with lots of data. However, the novelty of this work is limited. The use of 5-Fu containing MNs to treat abnormal scars had been reported by many groups. Some group even carried clinical trials using MNs for anti-scarring purpose. The separating MN concept was also widely reported. The only novel thing might be the synthesis of 5-Fu prodrug. Other issues are:

1. Figure 1: the mechanism of stimuli responsive MNs should be specified.
2. In the Efficient drug loading and responsive release performance of MN patches section, the relationship between drug release and therapeutic efficacy should be specified. Scar thicknesses varies, which affect the amount of drug needed.
3. Figure 3: The ROS response of 5-FuA-Pep-MA prodrug with different concentrations of H₂O₂ should be shown in both concentration and time.
4. The stability of the drug should be considered in the long term storage and humid environment.
5. In vivo therapeutic efficacy of separating MN patches on HS: both drug-loaded and drug-free MNs significantly reduced the levels of MMP2 and MMP9 in the HS pathological microenvironment, thereby inhibiting fibroblast migration. It is hard to conclude that the remodelling of the HS microenvironment was only associated with drug-loaded MNs. What is the mechanism for drug-free MNs that reduced the levels of MMP2 and MMP9 in the HS pathological microenvironment?

Reviewer #3:

Remarks to the Author:

The authors here present a comprehensive work on the novel establishment of drug-loaded microneedles to treat hypertrophic scars. In the reviewer's opinion, this study is of particular interest and provides a valuable new view on therapeutic options in hypertrophic scarring, as clinically, drug administration and sufficient skin permeation is often a problem in scar treatments, and highly effective drug delivery systems are scarce.

The authors provide a multi-model study and highlight the use of microneedles and their cellular effects in vitro and in vivo, in human and rabbit models.

Some minor concerns and questions arise when reading the paper:

- 1) Please indicate the models use either in the figure legends or include in the figure themselves, e.g. "Figure 6 Rabbit ear in vivo inhibition study..." to give easier orientation at a glance to the reader
- 2) Can you please elaborate on the translational aspects of the MN models: Can these drugloaded/separating MN patches be produced industrially and applied in larger scale? Do you plan on clinical safety and tolerability studies for the use of the 5-FU derivatives? Or are they already in use?
- 3) For better readability, maybe some abbreviations could be removed

Major concerns:

1) Neither from the manuscript text nor from the figures nor from the methods it can be read from which tissue/cells exactly the RNAseq is performed, I assume from the rabbit ear HS tissue...? Please indicate in the methods and in the figure legends. Did you use the full thickness skin/ear? Did you isolate the RNA from the biopsies, or isolate cells/dermis? Did you remove the epidermis?

2) Instead of RNAseq, single cell sequencing has been the gold standard for transcriptome investigations in the past few years. It is not even clear from what tissue/cells/material the RNAseq analysis was performed, however, if full-thickness rabbit ears were used, the RNAseq results are very unspecific. Complex processes such as hypertrophic scarring would need to be analyzed at higher resolution, at least separated by dermis and epidermis. Single-cell RNAseq would give much more detailed information about the interaction of the various cell types and their reaction to MN and 5-FU.

A point-by-point response to the reviewers' comments:

Reviewer #1:

This study developed a separating MN drug delivery system consisting of GelMA and 5-FuA prodrug in response to endogenous stimuli (MMP2, MMP9 and ROS) to remodel the pathological microenvironment for HS therapy. The novelty of this study was to develop a responsive MN system that prolonged the skin retention of 5-Fu, scavenged ROS and consumed MMPs for enhanced anti-scarring effect. However, the responsive drug release behavior was only confirmed from the *in vitro* experiments, and some important experiments are missing. Besides, therefore, there are still many other doubts and problems to be addressed. Therefore, I do not recommend it for acceptance by this journal. The detailed comments are shown as the following.

Response: Thanks for the comments. To respond, we have supplemented the following experiments: 1) The *in vivo* drug release behavior was assessed by *in vivo* imaging and histological analysis, which displayed slowly and linearly sustained drug release; 2) The structure of MN patches was characterized by fluorescence microscope to further elucidate the separating capacity of MN patches; 3) The drug release curve was retested by prolonging the incubation time till 120 h to verify the maximum drug release; 4) Intracellular ROS levels of the HSFb treated with different concentrations of 5-FuA-Pep-MA were analyzed to confirm the ROS scavenge ability of 5-FuA-Pep-MA; 5) Fluorescence and immunofluorescence staining with dihydroethidium (DHE) and 8-hydroxy-2'-deoxyguanosine (8-OHdG) were performed to demonstrate the ROS scavenging ability of drug-loaded MNs at the tissue staining level; 6) Single-cell RNA sequencing was performed to further understand the molecular mechanism and cell communication of MN treatment.

All the comments have been responded one-by-one in the following part. We believe that the revised manuscript is now suitable for publication in *Nature Communications*. Thanks for the comments and suggestions for further improvement of the manuscript.

1. The organization of introduction should be better improved. In addition to 5-Fu, intralesional injection of triamcinolone is the first-line therapy for HS. Why 5-Fu was selected for the treatment of HS? What is the advantage of MNs for drug delivery as compared to the injection. How the BAD and IGF1R pathways participate in the pathogenesis of HS? The related information was not provided, making the design of

this study not quite attractive.

Response: Thanks for the questions.

As the reviewer suggested, both 5-Fluorouracil (5-Fu) and corticosteroids (triamcinolone) are recommended therapies with different mechanisms in the treatment of HS (*J. Eur. Acad. Dermatol. Venereol.* 2021, 35, 2128; *J. Nippon. Med. Sch.* 2021, 88, 2). Corticosteroids could reduce inflammation response, promote vasoconstrictive effect, and inhibit fibroblast proliferation to treat HS. Although the use of corticosteroids to treat HS has been effective for most patients, it has also been associated with troublesome side effects of corticosteroids themselves, including hormonophobia, hypopigmentation, subcutaneous fat atrophy, telangiectasias, rebound effects and ineffectiveness.

5-Fu is capable of inhibiting cell proliferation, inducing fibroblast apoptosis, and decreasing collagen production, which acts as a therapeutic drug for HS treatment to avoid the potential side effects of corticosteroid injections. In clinical guideline and practice, 5-Fu has been used alone or in combination with corticosteroids to treat HS. (*Am. J. Clin. Dermatol.* 2023, 24, 225; *Front. Med. (Lausanne)* 2017, 4, 83.). Due to the short half-life (~10 to 20 min in blood circulation; *Oncol. Lett.* 2018, 15, 7175; *Cancer Res.* 1978, 38, 3479; *Cancer Biol. Ther.*, 2012, 13, 1407; *Cancer Chemother. Pharmacol.* 2013, 71, 351; *Biomater. Sci.* 2017, 5, 502), relatively high injection dose of 5-Fu and multiple operations are required. The high-dose local administration may induce serious side effects (*e.g.*, vasculitis, hyperpigmentation, erythema, purpura and burning sensation), and the invasive injection is accompanied with intense pain. Compared to triamcinolone, 5-Fu can be easily chemically modified to achieve a prolonged half-life for better therapeutic efficacy in the treatment of HS. Therefore, 5-Fu was selected for modification in this study. To respond, we have added the above description on Page 2, Line 13-25.

MNs have attracted significant interests as a new promising transdermal drug delivery platform. Compared to the intralesional injection, MNs have a lot of advantages, including nonselective loading capacity, minimal invasiveness, simple operation and good biocompatibility. Firstly, many therapeutic agents can be loaded

into MNs, including hydrophilic or hydrophobic small molecular drugs, and macromolecular drugs (proteins, mRNA, peptides, vaccines) for disease treatments. Secondly, MNs can penetrate through the stratum corneum layer of the skin into epidermis and dermis, creating microchannels on the skin to enhance transdermal drug delivery. More importantly, MNs can avoid contact with nerve fibers and blood vessels that reside primarily in the deep dermal layer. Therefore, MNs cause less pain than a hypodermic needle, leading to better patient compliance. Thirdly, intralesional injection technique requires the operation of an experienced physician or nurse. MNs can be self-administered at home. In addition, the thickness of the skin layers differs across individuals, the injection depth may be also varied. By adjusting the length of needles, MN could achieve all the layers of skin tissues to realize precisely drug release (*Front. Bioeng. Biotechnol.* 2022, 10, 1032041; *Micromachines (Basel)* 2021, 12, 1321). To respond, we have added the above description on Page 2, Line 25-28.

Increasing studies indicate that multiple pathways have been identified as relevant to the pathogenesis of fibrosis process and pathological scars, including inflammation pathways, anti-apoptosis pathways, growth factor pathways, angiogenesis pathways, etc (*Nature* 2020, 587, 555; *Nat. Rev. Rheumatol.* 2019, 15, 705; *Int. Wound J.*,2023, 20, 2190; *Ageing Res. Rev.* 2023, 83, 101809). In our study, bulk RNA sequencing (RNA-seq) analysis demonstrates that drug-loaded MNs could promote fibroblast apoptosis and inhibit collagen synthesis of HS tissues through down-regulation of BAD, IGF1R pathways and collagen fibril organization process. In gene set enrichment analysis (GSEA), the description of BAD pathway is regulation of BCL-2-associated death promoter (BAD) protein phosphorylation. Phosphorylation of the pro-apoptotic molecule BAD results in loss of the ability of BAD to heterodimerize with the survival proteins BCL-XL or BCL-2 inhibitors, thus inhibiting pro-apoptosis effect. Although BAD pathway is not reported in the pathogenesis of HS, it has been involved in the fibrotic process in other organs, such as lung and liver fibrosis (*Eur. J. Pharmacol.* 2014, 741, 281; *Gut.* 2006, 55, 1174; *Nat. Rev. Rheumatol.* 2020, 16, 11). The results of these studies support specific BCL-2 family pro-survival factors could promote myofibroblast survival to induce tissue fibrosis.

Insulin-like growth factor 1 receptor (IGF1R) is widely expressed in many cell types. Activation of the IGF-IR by binding its ligands IGF-1 and IGF-2 plays important roles in induction of proliferation, inhibition of cell apoptosis and induction of differentiation (*J. Mol. Endocrinol.* 2018, 61, 69). Up-regulated IGF-1/IGF-1R signaling is present in multiple fibrotic diseases, including idiopathic fibrotic lung diseases, secondary fibrosis and pathological scars (*Biomed. Res. Int.* 2018, 2018, 6057589; *Endocr. J.* 2000, 47, S41; *J. Invest. Dermatol.* 2000, 115, 1065-7; *Clin. Exp. Dermatol.* 2014, 39, 822). In addition, IGF-1R is highly expressed in dermal fibroblasts of HS and keloid (*Clin. Exp. Dermatol.* 2014, 39, 822), and IGF-1-induced AKT/mTOR/HIF-1 α signaling pathway is involved in the pathogenesis of keloid (*FASEB J.* 2023, 37, 23015.). Therefore, we speculate that BAD and IGF1R pathways play important roles in the fibrosis of rabbit ear HS model, and down-regulated BAD, IGF1R pathways of drug-loaded MNs treatment could promote fibroblast apoptosis and inhibit collagen synthesis of HS tissue.

To respond, we have added the above description on Page 2, Line 35-43; Page 3, Line 1-5; Page 21, Line 40-42 and Page 22, Line 1-8.

2. There are some grammatical errors in the manuscript, which need to be checked again and corrected.

e.g. The crosslinking degree could be controlled by UV light irradiation to endow the MN tips with tunable **mechanical** and drug release properties.

e.g. To visualize the drug distribution and penetration in local skin, the fluorescence microscopy images was taken **perpendicular** to skin surface at different depths using confocal laser scanning microscope (CLSM).

Response: Thanks for the suggestion. We have carefully corrected the typos in the revised manuscript.

3. There are some doubts that whether the mechanical strength of the separating MN is sufficient, since its mechanical strength is less than 50 N/patch in **Fig. 3a**. Besides, how the MNs achieved separation performance was not clearly provided. The advantage of separating MNs and UV-crosslinked MNs in improving the transdermal delivery efficiency was not described and related experiments were missing.

Response: Thanks for the comment. The mechanical strength of MNs can be effectively improved through photo-crosslinking. The fracture strength of MNs and the breaking force of a single tip could be analyzed according to the forced-displacement curve (**Fig.**

3a). In this study, the fracture force of a single tip with UV irradiation for 30, 45 and 60 s reached 0.80, 0.77 and 0.67 N/needle, respectively. Previous reports have shown that when the fracture force of a single needle is greater than 0.49 N/needle, MNs can penetrate HS tissues (*Bioact. Mater.* 2021, 6, 2400). Thus, it is suggested that the obtained MNs have enough mechanical strength to penetrate HS tissues.

To respond, we have added the above description on Page 6, Line 10-12. In the following *ex vivo* and *in vivo* experiments, MNs with the photo-crosslinking duration of 45 s were also demonstrated to penetrate HS tissues (**Fig. 4b**, **Fig. 4c**, **Supplementary Fig. 12** and **Supplementary Fig. 13**).

Fig. 3a Force curve of MN patches with different crosslinking time. Arrows indicate fracture force.

Fig. 4b Trypan blue staining of rabbit ear HS lesions (Left: HS lesion without treatment; Right: HS lesion treated with MNs). The scale bar in the last image can be applied to the others.

Fig. 4c Image of H&E-stained rabbit ear HS tissue after MN patch insertion.

Supplementary Fig. 12 Representative *in vivo* fluorescence microscopy images of mice at different time points treated with separating MN patches or integral MN patches, respectively.

Supplementary Fig. 13 Merged microscopy images of histological sections at different time points treated with separating MN patch. The scale bar in the last image can be applied to the others.

In this study, we prepared separating MN patches by a step-by-step perfusion method. Briefly, the tip precursor solution (including GelMA, 5-FuA-Pep-MA and photo-initiator) was added to the mould, and excess solution was then removed, leaving the precursor solution in the cavity. Notably, the tips were shrunk a bit during photo-crosslinking. Thus, when gelatin was added to the mould to construct the base layer, a small amount of gelatin entered the cavity to connect the tips. Namely, the MN tips

were mainly composed of photo-crosslinked GelMA, which could be slowly biodegraded in the body. In comparison, the base layer of MNs was composed of gelatin, which could be fast dissolved by the skin interstitial fluid after penetrating skin tissues. Specifically, the connection of MN tips with the base layer could be easily broken, leaving the tips in skin tissues, due to the presence of dissolvable gelatin in the bottom of MN tips. To visually demonstrate above speculation, FITC-labeled gelatin was used to prepare the base layer of MN patch, which was subjected to fluorescence microscope (**Supplementary Fig. 6**). After MNs penetrated through the stratum corneum layer of the skin, the connection of MN tips with the base layer was broken, leaving the bottom of the MN tips residue on the base layer. To make it clear, we have added above description on Page 5, Line 8-14.

Supplementary Fig. 6 Fluorescence microscopy image of MN patch with FITC-labeled gelatin as base layer.

In clinical practice, intralesional injection of 5-Fu as a rapid and effective manner is the priority choice. However, due to the elimination by endogenous dihydropyrimidine dehydrogenase, 5-Fu is greatly hindered by a very short half-time, approximately 10 to 20 min in blood circulation (*Oncol. Lett.* 2018, 15, 7175; *Cancer Res.* 1978, 38, 3479; *Cancer Biol. Ther.* 2012, 13, 1407; *Cancer Chemother. Pharmacol.* 2013, 71, 351; *Biomater. Sci.* 2017, 5, 502). Therefore, high injection dose of 5-Fu and multiple operations are required in clinical practice. The high-dose local administration may induce serious side effects (*e.g.*, skin atrophy, depigmentation and vasodilatation), and the invasive injection is accompanied with intense pain. In comparison, MNs as a new promising transdermal drug delivery platform have a lot of advantages, including

nonselective loading capacity, minimal invasiveness, self-administration and good biocompatibility. Specifically, separating MNs with photo-crosslinked MN tips undergo slow biodegradation in the skin, leading to long-term sustained release of encapsulated drugs. Therefore, MN patches consisting of FITC-Pep-MA were applied to mouse skin *in vivo*. Fluorescence images of the skin surface showed the dye release kinetics during MN biodegradation in the skin (**Supplementary Fig. 12**). Significant fluorescent spot corresponding to MN tips embedded in the skin could be observed initially, followed by gradual dimming over time. In comparison, when the integral MNs without separating capacity penetrated the skin, no fluorescence signal was observed in the skin of the mice, indicating negligible drug delivery. After MNs penetrated the skin, MN tips were fully embedded under the skin surface, as shown by histology (**Supplementary Fig. 13**). In addition, the photo-crosslinking strategy improved the mechanical strength of MN tips, leading to transdermal drug delivery in deeper tissues than dissolving MNs. The figure below showed the effective transdermal depth of non-crosslinked MNs, which is shorter than that of photo-crosslinked MNs (**Response letter Fig. 1**). Therefore, we believe that the separating MNs with photo-crosslinked tips have a more efficient drug delivery capability. We have added above description on Page 8, Line 8-16.

Response letter Fig. 1 CLSM images of porcine skin at varied depths after insertion of RhB-loaded MN (non-crosslinking) for 5 min. The scale bar in the last image can be applied to the others.

4. In **Fig. 3e** and **3h**, the degradation rate of GelMA and the drug release rate of 5-FuA

from the crosslinked GelMA/5-FuA-Pep-MA hydrogels with the UV-crosslinking duration of 45 s appeared to be approaching the plateau at 24 h, and it had better to continue studying how long it takes for the drug to achieve most release.

Response: Thanks for the suggestion.

Generally, MMPs enzyme activity can be only maintained for ~6 h in PBS and is almost completely disappeared at ~10 h. In previous experiments, the presence of the degradation and release plateau within 24 h should be attributed to the reduced enzyme activity, resulting in slower GelMA degradation and slower drug release. Here, we retested the degradation and release curves by supplementing MMP9 in the system every 12 h to maintain enzyme activity. Specifically, in the case of the degradation experiment, fresh PBS containing MMP9 replaced the previous solutions every 12 h, and the crosslinked GelMA hydrogel was taken out at a preset time point to record the wet weight for the calculation of the degradation rate (**Fig. 3e** and **Supplementary Fig. 8**). In the case of the drug release test, fresh MMP9 was added to the system every 12 h, and 50 μ L of solution was taken out at a preset time point, which was evaluated by HPLC to plot the drug release curve (**Fig. 3h**). On the 5th day, drug release plateau of the hydrogel with the photo-crosslinking duration of 45 s was achieved.

Fig. 3e Enzymatic degradation of GelMA with different crosslinking time. The data are presented as mean \pm SD (n = 3).

Supplementary Fig. 8 Effects of different enzyme activities on degradation rates of crosslinked GelMA. The data are presented as mean \pm SD (n = 3).

Fig. 3h Drug release curves of dual responsiveness drug delivery platform with different crosslinking time. Error bars represent the standard deviation (n = 3). *** $P < 0.001$.

5. Is there a contradiction that the separating MN was able to form microholes in rabbit skin for 5 days in Fig. 4d, but trans epidermal water loss decreased rapidly within 10 minutes in Fig. 4e.

Response: Thanks for the comment.

Transepidermal water loss (TEWL) is the most widely used objective measurement for assessing the barrier function of skin. It refers to the evaporation of water through epidermis or outer layer of skin. TEWL increases when the skin's protective barrier is damaged or impaired, allowing excess moisture content evaporate from the skin. When the separating MNs applied on the skin tissue, the mean TEWL value immediately increased from 8.6 to 10.8 and achieved peak value at 5 min. Subsequently, TEWL was gradually decreased to prior level within 60 min (**Fig. 4e**). In our opinion, the separating MNs damaged the epidermal barrier and increased TEWL

rapidly at first. However, as time increased, the tips of MNs separated from the base and entrapped in the skin. As shown in **Fig. 4d**, the regular arranged points inside the white dotted box were the tips of MNs entrapped in the skin. Due to the well-organized structure, elasticity, and tension of skin tissue, the tips of MN were enclosed tightly by surrounding skin tissue over time (**Response letter Fig. 2**). Therefore, water loss through epidermis was inhibited and the level of TEWL decreased. Due to the expressions of MMPs in skin tissue of rabbit ear, the crosslinked GelMA matrix was gradually degraded in skin tissue within 5 days. Therefore, the tip alignment formed by the separation of MNs (the regular arranged points inside the white dotted box) disappeared in rabbit skin for 5 days, which is not contradictory to the rapid decrease in TEWL within 10 min (*Exp. Dermatol.* 2005, 14, 386; *J. Invest. Dermatol.* 2018, 138, 2295; *ACS Appl. Mater. Interfaces* 2019, 11, 43588).

Response letter Fig. 2 Schematic diagram of the changes of skin tissue with MN tips insertion over time.

Fig. 4e TEWL from the MN-treated skins of five healthy volunteers before and after insertion of separating MN patches. Error bars represent the standard deviation (n = 5).

6. **Fig. 4b** and **4c** mainly showed that the separating MN has sufficient mechanical strength to break the skin barrier of HS tissues. It seems to need more evidence in the “*Penetration capacity and subcutaneous retention of separating MN patches*” section.

Response: Thanks for the suggestion.

We verified the separability and subcutaneous retention of MNs by histological analysis and *in vivo* fluorescence imaging. MN patches consisting of FITC-Pep-MA were applied to mouse skin *in vivo*. Fluorescence imaging of the skin surface showed the dye release kinetics during MN biodegradation in the skin (**Supplementary Fig. 12**). Significant fluorescent spot corresponding to MN tips embedded in the skin could be observed initially, followed by gradual dimming over time. After MNs penetrated the skin, MN tips were fully embedded under the skin surface, and the sizes of MN tips gradually decreased with extended time, as shown by histology (**Supplementary Fig. 13**). Notably, above results also demonstrated the long-term retention ability of the separating MNs. We have added above description on Page 8, Line 8-16.

Supplementary Fig. 12 Representative *in vivo* fluorescent images of mice at different time points treated with separating MN patches or integral MN patches, respectively.

Supplementary Fig. 13 Merged microscopy images of histological sections at different time points treated with separating MN patch. The scale bar in the last image can be applied to the others.

7. In **Fig. 6b**, the therapeutic effect of drug-loaded MNs and 5-Fu injection was not obvious, in comparison to drug-free MNs.

Response: Thanks for the comment.

Macroscopic images could show the surface, size and boundaries of rabbit ear scar. After the 3rd treatment and one-week follow-up, the macroscopic images showed that scars in the drug-loaded MNs group were smoother, and the scar boundaries were almost disappeared. Although a moderate therapeutic effect was obtained in 5-Fu injection group, newly formed needle-point scars were observed in the lesions. Drug-free MNs exhibited limited therapeutic efficacy, leading to the decreased height and blurred border of HS lesions. Due to the three-dimensional structure of scars, the two-dimensional information provided by macroscopic images cannot demonstrate the changes in scar height before and after treatments. Therefore, the therapeutic effects of different treatments were further evaluated by histopathology analysis and next genome sequencing.

Compared with macroscopic images, histopathology analysis could depict the amount, morphology, and arrangement of fibroblasts and collagen fibers. Scar elevation index (SEI) of histopathology analysis is an objective tool to evaluate the therapeutic effects in HS management, which refers to the ratio of the highest scar height by the lowest scar height to the surface of cartilage (*Bioact. Mater.* 2021, 6, 2400). As H&E staining images shown (**Fig. 6d** and **6e**), the thickness of HS tissue dramatically

decreased in drug-loaded MNs group after the 3rd administration. In comparison, HS tissue treated with 5-Fu injection displayed a bit thicker dermis and relatively disordered collagen distribution. The SEI value of the drug-loaded MNs group decreased from 3.5 to 1.4, while the HS group remained 3.3 ($P < 0.05$). Meanwhile, the SEI value of 5-Fu injection group was 2.2, which was significantly higher than that of the drug-loaded MNs group ($P < 0.05$). These results demonstrated that the drug-loaded MNs could effectively inhibit the fibroblasts over-proliferation and collagen synthesis, and exhibited better therapeutic effect than 5-Fu injection.

Fig. 6d,e H&E (d) and Masson's trichrome (e) staining of HS with different treatments before and after the 3rd evaluation. The scale bar in the last image can be applied to the others in the same panel.

8. The human hypertrophic scar fibroblast (HSFb) is usually adopted for the cellular study of HS, while NIH/3T3 fibroblasts was used to evaluate the ROS scavenge ability of 5-FuA-Pep-MA. I doubt whether this experiment is reasonable and 5-FuA-Pep-MA is available for scavenging ROS in HS tissues.

Response: Thanks for the comment.

We verified the ability of 5-FuA-Pep-MA to clear ROS not only in HSFb, but also in HS tissues by tissue staining. As shown in **Supplementary Fig. 15**, the fluorescence signal intensity in the experimental groups was significantly reduced with the increased concentration of 5-FuA-Pep-MA, confirming that 5-FuA-Pep-MA could serve as a ROS scavenger to efficiently remove the endogenous ROS in HSFb. To demonstrate the scavenging ability of drug-loaded MNs against ROS at the tissue staining level, DHE staining was performed to assess the ROS levels in the pathological microenvironment of HS with different treatments (**Fig. 6f**). The results of DHE staining showed that drug-loaded MNs could significantly reduce ROS levels in tissues compared to HS, drug-free MNs and 5-Fu injection groups. Meanwhile, the fluorescence intensities of 8-OHdG staining in green channel showed that drug-loaded MNs could significantly relieve the oxidative damage caused by ROS. We have added above description on Page 10, Line 18-23 and Page 12, Line 31-35.

Supplementary Fig. 15 Intracellular ROS levels of the HSFb treated with different concentrations of 5-FuA-Pep-MA. The scale bar in the last image applies to the others.

Fig. 6f Fluorescence and immunofluorescence microscopy images staining for DHE, 8-OHdG, MMP2 and MMP9 in NS and HS tissues with different treatments. The scale bar in the last image can be applied to the others.

9. The reference format is inconsistent, and it had better to refer to more recent studies.

Response: Thanks for the suggestion. We have revised to address this issue.

10. The resolution of Fig.3 and Fig. 6-8 is poor.

Response: Thanks for the suggestion. We've uploaded images with high resolution.

11. The first appearance of BAD and IGF1R in the abstract and introduction should provide the full name.

Response: Thanks for the suggestion. We have already provided the full names of BAD when their appearance.

Reviewer #2:

This study reports the use of a separating microneedle system for sustained 5-fluorouracil (5-Fu) in the hypertrophic scar (HS) treatment. 5-Fu was designed as a prodrug through binding with ROS-responsive tetrapeptide (PPPK) at the N-terminus and methacryloyl on the lysine residue, which was UV-crosslinked with GelMA as the tips of separating MN patches. Later the detached MN tips slowly released 5-Fu, which reduced abnormal fibroblast proliferation and collagen fiber deposition.

In general, the experiments were designed well with lots of data. However, the novelty of this work is limited. The use of 5-Fu containing MNs to treat abnormal scars had been reported by many groups. Some group even carried clinical trials using MNs for anti-scarring purpose. The separating MN concept was also widely reported. The only novel thing might be the synthesis of 5-Fu prodrug.

Response: Thank you for the comments.

We respectively disagree with the comment that the novelty of this work is limited. We believe that the novelty of this work lies in the design of a peptide prodrug with ROS response property, the development of a multi-responsive MNs system, and provide a multi-model study to highlight the use of MNs and their therapeutic effects *in vivo*. Specifically, we developed a separating MN drug delivery system consisting of GelMA and 5-FuA-Pep-MA prodrug in response to endogenous stimuli (MMP2, MMP9 and ROS) to remodel the pathological microenvironment for HS treatment. The separating MN drug delivery system prolonged the focal retention of 5-Fu, increased

the half-life of the drug, and improved the bioavailability of the drug. Moreover, it can remodel the pathological microenvironment by scavenging ROS and depleting MMPs to enhance the anti-scar effect.

More importantly, the unique therapeutic mechanism of biomaterials on HS treatments is usually neglected in biomaterial science, which is very important for the following basic research and clinical translation. Bulk RNA sequencing analysis demonstrates that drug-loaded MNs could promote fibroblast apoptosis and inhibit collagen synthesis through down-regulation of BCL-2-associated death promoter (BAD), insulin-like growth factor 1 receptor (IGF1R) pathways and collagen fibril organization process, simultaneously stimulating inflammatory response, and mediating keratinocyte differentiation via up-regulation of toll-like receptors (TOLL), interleukin-1 receptor (IL1R) and keratinocyte pathways. Single-cell RNA sequencing (scRNA-seq) is a new technology for sequencing transcriptome at the single cell level, which can study gene expression within a single cell and investigate the intercellular communication. In our study, scRNA-seq further suggested that the interactions between fibroblast and keratinocyte via ligand-receptor pair of proteoglycans 2 (HSPG2)-dystroglycan 1 (DAG1), played a central role in treating HS tissue with drug-loaded MNs. This study revealed the potential therapeutic mechanism of drug-load MNs.

All the specific comments were responded one by one in the following part.

1. Figure 1: the mechanism of stimuli responsive MNs should be specified.

Response: According to the suggestion, we have revised **Fig. 1**, in which we added the text description of MNs degradation (MMPs-degraded GelMA matrix) and marked the ROS response site of 5-FuA-Pep-MA prodrug (green scissors).

Specifically, GelMA could be degraded by overexpressed MMPs in the pathological microenvironment of HS. Meanwhile, the high ROS levels could cleave the PPP tripeptide sequence of 5-FuA-Pep-MA. Therefore, MN tips consisting of photo-crosslinked GelMA and 5-FuA-Pep-MA can be biodegraded in response to both MMPs and ROS. Notably, the drug-loaded MNs in response to endogenous stimuli

(MMP2, MMP9 and ROS) could decrease the ROS level and consume MMP2 and MMP9, leading to remodel the pathological microenvironment of HS tissues and achieve a better therapeutic effect.

Fig. 1 Schematic diagram of the separating MN patch as a pathological microenvironment-responsive peptide prodrug delivery platform.

2. In the Efficient drug loading and responsive release performance of MN patches section, the relationship between drug release and therapeutic efficacy should be specified. Scar thicknesses varies, which affect the amount of drug needed.

Response: Thanks for the comment. We prepared ROS- and MMPs-responsive separating MN patches consisting of photo-crosslinked GelMA and 5-FuA-Pep-MA prodrug. The responsiveness properties of MNs could not only slowly release 5-Fu derivative, but also consume ROS and MMPs to remodel the pathological microenvironment of HS tissues. *In vitro* drug release experiment indicated that, comparing drug release rates only, MNs released drugs slowly in response to endogenous stimuli, extending the drug release period to about a week. In addition, the bioavailability of drug molecules was greatly improved due to the slowing down of the release rate. Moreover, *in vivo* HS inhibition experiment demonstrated that the drug-loaded MNs could effectively inhibit over-proliferation of fibroblasts and collagen synthesis, exhibiting better therapeutic effect than 5-Fu injection, even if the dose of MNs is reduced to half of the injection. Notably, the levels of ROS and MMPs in the

pathological microenvironment of HS tissues determined the drug release rate and duration.

Indeed, the size, thickness and duration vary in patients with HS, which could affect the amount of drug needed. The higher the scar thickness, the more active fibroblasts/myofibroblasts and the more collagen fibers secreted. Therefore, we could adjust the frequency of treatment for different scar features (size, thickness and duration).

3. Figure 3: The ROS response of 5-FuA-Pep-MA prodrug with different concentrations of H₂O₂ should be shown in both concentration and time.

Response: Thanks for the suggestion. We have revisited the relevant experiment. Specifically, 5-FuA-Pep-MA prodrug was incubated with PBS containing different concentrations (10, 50, 100 and 300 μM) of H₂O₂ at 37 °C, respectively. 50 μL solution was taken out at a preset time (24, 48, 72, 96 and 120 h), which was analyzed by HPLC. The results indicated that the response of 5-FuA-Pep-MA prodrug to H₂O₂ was time- and concentration-dependent (**Fig. 3g**).

Fig. 3g ROS response of 5-FuA-Pep-MA prodrug with different concentrations of H₂O₂. Error bars represent the standard deviation (n = 3).

4. The stability of the drug should be considered in the long term storage and humid environment.

Response: Thanks for the suggestion. 5-Fu is a relatively stable small molecule drug, which is usually stored as an aqueous solution at room temperature for a long time in clinical application. Peptides also have good stability in a sterile and enzyme-free environment. Therefore, we believe that the drug has good stability during long-term storage in MN patches. However, the storage of MNs in high humid environment will

reduce the penetrating capacity of MN tips due to the hydrogel swelling, even leading to the collapse of the tip structure. To assess the stability of the drug, drug-loaded MNs stored at an electronic moisture-proof box (room temperature and 50% relative humidity) for 9 months were applied for the treatment in rabbit ear HS model. The results showed that drug-loaded MNs still had excellent biological effects after long-term storage, as well as good therapeutic effects on HS (**Supplementary Fig. 17**).

To respond, we have added above description on Page 12, Line 42-44 and Page 13, Line 1.

Supplementary Fig. 17 Evaluation of *in vivo* therapeutic effect on HS with drug-loaded MN patch after long-term storage. a, Representative photographs of HS before and after different treatments. b, SEI of different groups after the 3rd evaluation. c, H&E staining of HS with different treatments before and after the 3rd evaluation. d, Masson's trichrome staining of HS with different treatments before and after the 3rd evaluation. The scale bar in the last image can be applied to the others in the same panel. Error bars represent the standard deviation (n = 3). ns means no significance, **P* < 0.05.

5. *In vivo* therapeutic efficacy of separating MN patches on HS: both drug-loaded and drug-free MNs significantly reduced the levels of MMP2 and MMP9 in the HS pathological microenvironment, thereby inhibiting fibroblast migration. It is hard to conclude that the remodelling of the HS microenvironment was only associated with drug-loaded MNs. What is the mechanism for drug-free MNs that reduced the levels of MMP2 and MMP9 in the HS pathological microenvironment?

Response: Thanks for the comment. It has been reported that MMPs were found to be highly expressed in the pathological microenvironment of HS compared to normal skin

tissue. (*J. Plast. Reconstr. Aesthet. Surg.* 2010, 63, 1015) GelMA is a directional modification of gelatin, which has cell adhesion sites and degradation sites for MMPs (*Dent. Mater.* 2018, 34, 389; *ACS Appl. Mater. Interfaces* 2020, 12, 16006). The results of immunohistochemical and immunofluorescence analysis verified that both drug-loaded and drug-free MNs significantly reduced the levels of MMP2 and MMP9 in the HS pathological microenvironment (**Fig. 6f** and **Supplementary Fig.16**). However, the remodelling of the HS microenvironment was associated with not only the reduction of the levels of MMP2 and MMP9, but also the reduction of ROS levels. To demonstrate the scavenging ability of drug-loaded MNs against ROS at the tissue level, dihydroethidium (DHE) staining was performed to assess the ROS levels in the pathological microenvironment of HS with different treatments (**Fig. 6f**). The results of DHE staining showed that drug-loaded MNs could significantly reduce ROS levels in tissues compared to HS, drug-free MNs and 5-Fu injection groups. Meanwhile, the fluorescence intensities of 8-hydroxy-2'-deoxyguanosine (8-OHdG) staining in green channel showed that drug-loaded MNs could significantly relieve the oxidative damage caused by ROS. In addition, the toxicity of 5-Fu derivative against fibroblasts also led to the reduction of the levels of MMPs and ROS secreted by fibroblasts in HS tissues. Therefore, we believe that drug-loaded MNs could induce the remodelling of the HS microenvironment.

Fig. 6f Fluorescence and immunofluorescence microscopy images staining for DHE, 8-OHdG, MMP2 and MMP9 in NS and HS tissues under different treatment groups, respectively. The scale bar in the last image can be applied to the others.

Supplementary Fig. 16 Immunohistochemistry staining for MMP2 (first column) and MMP9 (second column) in NS and HS tissues under different treatments. The scale bar in the last image can be applied to the others.

As for the mechanism of drug-free MNs reducing the levels of MMP2 and MMP9 in the HS pathological microenvironment, on the one hand, GelMA in drug-free MNs could consume MMP2 and MMP9; on the other hand, drug-free MNs could also remodel hypertrophic scars *via* a mechanical communication pathway (*Cell Death Dis.* 2021, 12, 226; *Front. Immunol.* 2022, 13, 1028410; *ACS Nano* 2022, 16, 10163). Therefore, the results of immunohistochemical and immunofluorescence analysis verified that drug-free MNs could also reduce the levels of MMP2 and MMP9 in the HS pathological microenvironment. Notably, immunofluorescence analysis of drug-loaded MNs showed the lower levels of MMP2 and MMP9 (red channel) than that of drug-free MNs, indicating better therapeutic effect of drug-loaded MNs on HS.

Reviewer #3:

The authors here present a comprehensive work on the novel establishment of drug-loaded microneedles to treat hypertrophic scars. In the reviewer's opinion, this study is of particular interest and provides a valuable new view on therapeutic options in hypertrophic scarring, as clinically, drug administration and sufficient skin permeation

is often a problem in scar treatments, and highly effective drug delivery systems are scarce. The authors provide a multi-model study and highlight the use of microneedles and their cellular effects in vitro and in vivo, in human and rabbit models.

Response: Thanks for the positive evaluation and valuable suggestions for further improvement of our manuscript. We supplemented single-cell RNA sequencing (scRNA-seq) to further analyze the interactions between keratinocytes and fibroblasts under drug-loaded MNs treatment, which could analyze complex functions of drug-loaded MNs in treating HS at higher resolution. In addition, all the specific comments were responded one by one in the following part.

Minor concerns:

1) Please indicate the models use either in the figure legends or include in the figure themselves, e.g., “Figure 6 Rabbit ear in vivo inhibition study...” to give easier orientation at a glance to the reader.

Response: Thanks for the suggestion. We have already modified the figure caption of **Fig. 4, 6 and S11**.

2) Can you please elaborate on the translational aspects of the MN models: Can these drugloaded/separating MN patches be produced industrially and applied in larger scale? Do you plan on clinical safety and tolerability studies for the use of the 5-FU derivatives? Or are they already in use?

Response: Thanks for the comment. Due to the simplicity of the preparation process, the drug-loaded separating MNs could be manufactured industrially and in larger scale. We have already had the capability of large-scale industrial production of drug-loaded/separating MN patches. Currently, the drug-loaded separating MNs are performed in animal experiments. Before we conducted preclinical trials on human subjects, the safety and therapeutic efficacy studies of 5-Fu derivatives would be strictly investigated in larger groups of healthy subjects and patients. The first MN clinical trial (NCT00558649) was conducted in May 2007 to compare the effectiveness and safety of administering low-dose influenza vaccine intradermally. Subsequently, the number of MN clinical trials increased significantly in 2012 and has steadily increased. Dissolving MN patches are fabricated using hydrophilic materials. It has been used to treat sweat induction, infraorbital wrinkle (NCT04989361), psoriasis (NCT02955576), solar lentigines (NCT04583852), basal cell carcinoma (NCT04928222 and NCT03646188), cutaneous T cell lymphoma (NCT02192021), cutaneous squamous

cell carcinoma (NCT05377905) and local pain (NCT05078463) (*J. Control. Release* 2023, 360, 687).

In our study, ROS- and MMPs-responsive separating MN consisting of gelatin methacryloyl. It has been reported that gelatin MNs have been used in clinical trials. Rouphael *et al.*, have shown in phase 1 clinical trials that their gelatin-based MNs could produce a robust antibody response towards an influenza vaccine (*J. Control. Release* 2022, 348, 186; *Lancet* 2017, 390, 649).

3) For better readability, maybe some abbreviations could be removed.

Response: Thanks for the suggestion. We have revised to address this issue.

Major concerns:

1) Neither from the manuscript text nor from the figures nor from the methods it can be read from which tissue/cells exactly the RNAseq is performed, I assume from the rabbit ear HS tissue...? Please indicate in the methods and in the figure legends. Did you use the full thickness skin/ear? Did you isolate the RNA from the biopsies, or isolate cells/dermis? Did you remove the epidermis?

Response: Thanks for the comment. In the *in vivo* experiment of this study, 15 rabbits were randomly assigned into 5 groups, including control group (group 1, NS), group 2 (HS without treatment), group 3 (HS with drug-free MNs), group 4 (HS with drug-loaded MNs), and group 5 (HS with 5-Fu injection). On day 43rd (one week after the 3rd administration), three samples were collected randomly in group 1, 2, 4 and 5 for bulk RNA sequencing (RNA-seq). To explore the effects of different treatments on all levels of skin tissues, full thickness scar tissues of rabbit ears was used to isolate the RNA. All 12 samples were subjected to RNA extraction using RNA Isolation Kit. Subsequently, RNA integrity was assessed using the RNA Nano 6000 Assay Kit of the Bioanalyzer 2100 system (Agilent Technologies, CA, USA). High throughput sequencing was completed with an Illumina NovaSeq 6000 System (Illumina Inc., San Diego, CA).

To respond, we have added above description in Supplementary information.

2) Instead of RNAseq, single cell sequencing has been the gold standard for transcriptome investigations in the past few years. It is not even clear from what tissue/cells/material the RNAseq analysis was performed, however, if full-thickness rabbit ears were used, the RNAseq-results are very unspecific. Complex processes such

as hypertrophic scarring would need to be analyzed at higher resolution, at least separated by dermis and epidermis. Single-cell RNAseq would give much more detailed information about the interaction of the various cell types and their reaction to MN and 5-FU.

Response: Thanks for the comment and suggestion. Bulk RNA-seq displays the average expression level of a single gene in cell populations, which can be used to explore differences in gene expression between different conditions (such as different treatments). Therefore, bulk RNA-seq analysis of full thickness scar tissues of rabbit ears were used to explore the effects of different treatments on all levels of HS tissues. ScRNA-seq is a new technology for sequencing transcriptome at the single cell level. It can study gene expression within a single cell and solve the problem of cell heterogeneity.

To respond, we have added the results of scRNA-seq on isolated epidermis and dermis of rabbit ear scar model to further analyze the interactions between keratinocytes and fibroblasts under drug-loaded MNs treatment, which could analyze complex functions of drug-loaded MNs in treating HS at higher resolution. ScRNA-seq analysis further suggested that the interactions between fibroblasts and keratinocytes via ligand-receptor pair of proteoglycans 2 (HSPG2)-dystroglycan 1(DAG1), played a central role in treating HS tissue with drug-loaded MNs. The results of scRNA-seq have added in the manuscript (**Fig. 9**) on Page 3, Line 22-25; Page 18, Line 1-43; Page 19, Line 1-15; Page 22, Line 20-32 and Page 22, Line 39-43.

Fig. 9 Single-cell RNA-seq of HS tissues with no treatment and drug-loaded-MNs treatment. **a**, Schematic representation of scRNA-seq procedure in epidermis and dermis of rabbit ear HS tissue. **b**, Uniform manifold approximation and projection

(UMAP) plot of the profiled single cells. Cells with the same colors are from the same group. **c**, UMAP clustering of all the cells by classic markers and particular transcriptional signatures. **d**, UMAP plot of all the cells annotated into 9 major cell types. **e**, Bar plots of significantly enriched GO terms of the DEGs in keratinocytes. **f**, Bar plots of significantly enriched GO terms of the DEGs in fibroblasts. **g**, Cell-cell communications between any two identified cell types illustrated with the interaction numbers for each of the cell types in drug-loaded MNs group. **h**, Cell-cell communications between any two identified cell types illustrated with the interaction strength for each of the cell types in drug-loaded MNs group. **i**, The significantly ligand-receptor pairs of mostly communicated cell types in control and drug-loaded MNs groups. **j**, Bubble plot of the expression level of HSPG2-DAG1 pair among different cell types in drug-loaded MNs group

Reviewers' Comments:

Reviewer #1:

Remarks to the Author:

The authors have carefully revised all the comments raised by the reviewer. I think the revised manuscript is now acceptable for publication in this journal.

Reviewer #2:

Remarks to the Author:

This revised manuscript is much better than the original version. However, the following should be addressed.

1. Fig. 6b, the images are too fuzzy and can't distinguish the difference between the drug-free MN and drug-load MN in days 50. Also, fig. 7, fig. 9 and fig. 8d are fuzzy and the font size is too small.
2. The figures in fig. 3 need to be aligned better.
3. This work claimed an innovation part that is ROS responsive release of 5-FUA from prodrug. Which data directly presents this responsibility? I think the S fig.15 can't show the relationship between ROS level and release rate. It is important to present the responsibility, and prove that the MN would not result in high-dose release, side effects in scar lesions.

Reviewer #3:

Remarks to the Author:

The authors have thoroughly reworked all reviewer remarks and significantly improved the manuscript.

I have only a few more comments:

In Figure 7 and 8, the samples from which the gene analysis was performed are still not mentioned in the figure legend. Please state in the figure legend, e.g. "The gene distribution and functional annotation analysis in full thickness rabbit HS ear model" and also mention in the manuscript text, e.g. in line 353: "To elucidate the RNA changes associated with different treatments on HS, RNA-seq analysis was performed from full thickness rabbit HS model" to improve orientation within the article.

The authors have extended their analyses from bulk to single-cell transcriptomics, which is greatly appreciated. However, the differences of activity in receptor-ligand pairs (Figure 9i) are minimal, and the figure is hardly readable. Either reduce the number of interactions shown on the graph, or maybe a different graphic to illustrate the differential activities MN vs control would be more advantageous, e.g. the top 50 differential expressed genes in FBs and KCs.

A point-by-point response to the reviewers' comments:

Reviewer #2:

This revised manuscript is much better than the original version. However, the following should be addressed.

1. Fig. 6b, the images are too fuzzy and can't distinguish the difference between the drug-free MN and drug-load MN in days 50. Also, fig. 7, fig. 9 and fig. 8d are fuzzy and the font size is too small.

Response: Thanks for the suggestion.

To respond, we have revised the **Fig. 6b** to distinguish the difference between the drug-free MNs and drug-loaded MNs groups in day 50th by adjusting the contrast (drug-free MNs group) and choosing the most representative photograph (drug-loaded MNs group). The macroscopic images showed that scars in the drug-loaded MNs group were flattened, and the scar boundaries were almost disappeared in day 50th, while limited therapeutic efficacy of drug-free MNs treatment resulted in more distinct bulge and blurred border in HS lesions. Macroscopic images could show the surface, size and boundaries of rabbit ear scars, but fail to display the height of scars. Therefore, scar elevation index (SEI) of histopathology analysis was performed to further quantitatively evaluate the therapeutic effects in HS management (**Fig. 6c**). The SEI values demonstrated that the drug-loaded MNs could effectively decrease the height of scars, and exhibited better therapeutic effect than drug-free MNs.

In addition, we have rearranged **Fig.7, 8 and 9**. Besides, we uploaded the original images of **Fig. 7, 8 and 9** as attachments, since the compression of the submission system might make the images blurred.

Fig. 6 Evaluation of *in vivo* therapeutic effect on rabbit ear HS under different interventions and reconstruction of pathological microenvironment. a, Diagram of the establishment of HS model and evaluation of the therapeutic effect. **b**, Representative photographs of HS before and after different treatments. **c**, SEI of different groups after the 3rd evaluation. **d**, H&E staining of HS with different treatments before and after the 3rd evaluation. **e**, Masson's trichrome staining of HS with different treatments before and after the 3rd evaluation. **f**, Fluorescence and immunofluorescence staining for DHE, 8-OHdG, MMP2 and MMP9 in NS and HS tissues with different treatments. The scale bar in the last images can be applied to the

others in the same panel. Error bars represent the standard deviation (n = 3). ns means no significance, * $P < 0.05$.

Fig. 7 The gene distribution and functional annotation analysis in full thickness rabbit ear HS model. **a**, Global gene expression distribution across all samples. **b**, 2D PCA plot of the relationships among all samples. **c**, Three-dimensional PCA plot indicating the separation of all samples into 4 clusters. **d**, Barplot of DEGs between different groups. **e**, Heatmap of DEGs among NS, HS, drug-loaded MNs and 5-Fu injection groups. **f**, Volcano plot for DEGs between HS and NS groups. The green plots represent down-regulated genes, and red plots indicate up-regulated genes. **g**, Volcano plot between drug-loaded MNs and HS groups. **h**, Volcano plot between 5-Fu injection and HS groups. **i**, Volcano plot between drug-loaded MNs and 5-Fu injection groups. **j**, The significantly enriched BP of GO terms between HS and NS groups. **k**, The significantly enriched CC of GO terms between HS and NS groups. **l**, The significantly enriched MF of GO terms between HS and NS groups. **m**, The most significantly enriched KEGG signaling pathways between HS and NS groups.

Fig. 8 The function annotation of DEGs in full thickness rabbit ear HS model with drug-loaded MNs and 5-Fu injection treatment. **a**, Venn diagram showed the intersection of DEGs obtained by drug-loaded MNs compared with HS and 5-Fu injection compared with HS analysis. **b**, GSEA results for up-regulated and down-regulated GO terms compared between drug-loaded MNs and HS groups. **c**, GSEA results for up-regulated and down-regulated GO terms compared between 5-Fu

injection and HS groups. **d**, GSEA enrichment plot showing the top ranked down-regulated and up-regulated BioCarta pathways between drug-loaded MNs and HS groups. **e**, GSEA enrichment plot showing the top ranked down-regulated and up-regulated BioCarta pathways between 5-Fu injection and HS groups. **f**, The subnetwork of the 12 significant genes with higher degrees in the PPI network selected by MCODE. **g**, Vertical bars of upper plot showing the intersection of the 20 hub genes screened by six algorithms in cytoHubba.

Fig. 9 Single-cell RNA-seq of epidermis and dermis in rabbit ear HS model with no treatment and drug-loaded MNs treatment. **a**, Schematic representation of scRNA-seq procedure in epidermis and dermis of rabbit ear HS tissue. **b**, UMAP plot of the profiled single cells. Cells with the same colors are from the same group. **c**, UMAP clustering of all the cells by classic markers and particular transcriptional signatures. **d**, UMAP plot of all the cells annotated into 9 major cell types. **e**, Bar plots of significantly enriched GO terms of the DEGs in keratinocytes. **f**, Bar plots of significantly enriched GO terms of the DEGs in fibroblasts. **g**, Cell-cell communications between any two identified cell types illustrated with the interaction numbers for each of the cell types in drug-loaded MNs group. **h**, Cell-cell communications between any two identified cell types illustrated with the interaction strength for each of the cell types in drug-loaded MNs group. **i**, The significantly ligand-receptor pairs of mostly communicated cell types in control and drug-loaded MNs groups. **j**, Bubble plot of the expression level of HSPG2-DAG1 pair among different cell types in drug-loaded MNs group.

2. The figures in fig. 3 need to be aligned better.

Response: Thanks for the suggestion. We have rearranged the **Fig. 3**.

Fig. 3 Characterization, efficient drug loading and responsive drug release of MN patches. **a**, Force curve of MN patches with different crosslinking time. Arrows indicate fracture force. **b**, Swelling ratios of MNs with different crosslinking time. **c**, Drug loading amount of 5-FuA prepared from various concentrations of drug loading solutions. **d**, Crosslinking efficiency of MNs with different crosslinking time. **e**, Enzymatic degradation of GelMA with different crosslinking time. **f**, SEM images of MN patches before (left) and after (right) enzymatic degradation. **g**, ROS response of 5-FuA-Pep-MA prodrug with different concentrations of H₂O₂. **h**, Drug release curves of dual responsiveness drug delivery platform with different crosslinking time. Error bars represent the standard deviation (n = 3). ****P* < 0.001.

3. This work claimed an innovation part that is ROS responsive release of 5-FUA from prodrug. Which data directly presents this responsibility? I think the S fig.15 can't show

the relationship between ROS level and release rate. It is important to present the responsibility, and prove that the MN would not result in high-dose release, side effects in scar lesions.

Response: Thanks for the comments and the question.

We designed a peptide prodrug with ROS response property using a tripeptide (-Pro-Pro-Pro-) sequence with ROS response performance, as shown in **Fig. 3g**. HPLC test showed that the responsive release of prodrug was positively correlated with ROS concentration and incubation time. In addition, the ROS cleavage of the prodrug was studied by LC-MS (**Supplementary Fig. 9**). The toxicity of the released drug and the ability to inhibit cell migration were verified in cell experiments (**Fig. 5**). The results showed that the peptide prodrug 5-FuA-Pep-MA could be cleaved to 5-FuA-Pro-Pro in the presence of ROS, which had the similar cytotoxicity and inhibition of cell migration with 5-Fu. As for the relationship between ROS level and release rate, we plotted the drug release curves under different ROS concentrations by HPLC *in vitro* (**Fig. 3g**), indicating that ROS level and release rate had a concentration and time dependent relationship.

In this work, we designed a MN drug delivery system in response to overexpressed substances (ROS and MMPs) in the HS pathological microenvironment. On the one hand, the response rate of the peptide prodrug was related to the level of ROS and the incubation time, and the response behavior was a slow and continuous process (**Fig. 3g**). Moreover, **Supplementary Fig. 15** displayed that the presence of 5-FuA-Pep-MA could reduce the intracellular ROS level of HSFb, suggesting that the prodrug could undergo the ROS cleavage and drug release at the cellular level. On the other hand, *in vitro* experiments showed that GelMA, the main component of MNs, was degraded slowly by MMP9 (**Supplementary Fig. 8**), and its degradation rate could be regulated by the photo-crosslinking time (**Fig. 3e**). Moreover, the *in vivo* degradation and drug release of MNs could be monitored by fluorescence imaging and tissue images (**Fig. 4d, Supplementary Fig. 12 and 13**). This dual-response design resulted in a slow and continuous release of the peptide prodrug (**Fig. 3h**).

In summary, these results demonstrated that the drug release in scar lesions should

be slow and sustained, indicating that the MN would not result in high-dose release, side effects in scar lesions.

Reviewer #3:

The authors have thoroughly reworked all reviewer remarks and significantly improved the manuscript.

I have only a few more comments:

In Figure 7 and 8, the samples from which the gene analysis was performed are still not mentioned in the figure legend. Please state in the figure legend, e.g. “The gene distribution and functional annotation analysis in full thickness rabbit HS ear model” and also mention in the manuscript text, e.g. in line 353: “To elucidate the RNA changes associated with different treatments on HS, RNA-seq analysis was performed from full thickness rabbit HS model” to improve orientation within the article.

Response: Thanks for the suggestions.

We have already modified the figure legends of **Fig. 7, 8 and 9**. According the suggestion, we have added the above description on Page 14, Line 3.

The authors have extended their analyses from bulk to single-cell transcriptomics, which is greatly appreciated. However, the differences of activity in receptor-ligand pairs (Figure 9i) are minimal, and the figure is hardly readable. Either reduce the number of interactions shown on the graph, or maybe a different graphic to illustrate the differential activities MN vs control would be more advantageous, e.g. the top 50 differential expressed genes in FBs and KCs.

Response: Thanks for the suggestions.

We have reduced the number of intercellular interactions in **Fig. 9i** to make it more readable. To fully illustrate the mechanism of drug-loaded MNs, the top 50 DEGs in keratinocytes and fibroblasts between drug-loaded MNs and HS groups were displayed in heatmaps, respectively (**Supplementary Fig. 25 and 26**).

To respond, we have added the above description on Page 18, Line 31-33.

Fig. 9 Single-cell RNA-seq of epidermis and dermis in rabbit ear HS model with no treatment and drug-loaded MNs treatment. **a**, Schematic representation of scRNA-seq procedure in epidermis and dermis of rabbit ear HS tissue. **b**, UMAP plot of the profiled single cells. Cells with the same colors are from the same group. **c**, UMAP clustering of all the cells by classic markers and particular transcriptional signatures. **d**, UMAP plot of all the cells annotated into 9 major cell types. **e**, Bar plots of significantly enriched GO terms of the DEGs in keratinocytes. **f**, Bar plots of

significantly enriched GO terms of the DEGs in fibroblasts. **g**, Cell-cell communications between any two identified cell types illustrated with the interaction numbers for each of the cell types in drug-loaded MNs group. **h**, Cell-cell communications between any two identified cell types illustrated with the interaction strength for each of the cell types in drug-loaded MNs group. **i**, The significantly ligand-receptor pairs of mostly communicated cell types in control and drug-loaded MNs groups. **j**, Bubble plot of the expression level of HSPG2-DAG1 pair among different cell types in drug-loaded MNs group.

Supplementary Fig. 25 DEGs in keratinocytes. The heatmap shows the top 50 DEGs in keratinocytes between drug-loaded MNs and HS groups.

Supplementary Fig. 26 DEGs in fibroblasts. The heatmap shows the top 50 DEGs in fibroblasts between drug-loaded MNs and HS groups.

Reviewers' Comments:

Reviewer #2:

Remarks to the Author:

The authors have carefully revised all the comments raised by the reviewer. Some minor changes are still required.

For the question3: As described in method: "The peak area obtained from the sample without H₂O₂ was recorded as A₀, while the peak area obtained from the sample with H₂O₂ was recorded as A_c. The degradation rate was calculated by $[(A_0 - A_c)/A_0] \times 100\%$ to evaluate the ROS response of peptide prodrug"

I think the peak area (represents the prodrug) changing rate indicates the degradation, but is not equal to the release of 5-FU yet.

A direct observation of more 5-FU with more H₂O₂, by HPLC or other methods, either in-vitro or in-vivo, would be more convictive. For example, glucose responsive release of insulin was detected by insulin elisa kit, not the degradation of vesicles (10.1073/pnas.1505405112).

Reviewer #3:

Remarks to the Author:

The authors have answered my requests satisfactorly and significantly improved the figures and the manuscript. I suggest acceptance for publication.

A point-by-point response to the reviewers' comments:

Reviewer #2:

The authors have carefully revised all the comments raised by the reviewer. Some minor changes are still required.

For the question3: As described in method: The peak area obtained from the sample without H₂O₂ was recorded as A₀, while the peak area obtained from the sample with H₂O₂ was recorded as A_c. The degradation rate was calculated by $[(A_0 - A_c)/A_0] \times 100\%$ to evaluate the ROS response of peptide prodrug.

I think the peak area (represents the prodrug) changing rate indicates the degradation, but is not equal to the release of 5-FU yet.

A direct observation of more 5-FU with more H₂O₂, by HPLC or other methods, either in-vitro or in-vivo, would be more convictive. For example, glucose responsive release of insulin was detected by insulin elisa kit, not the degradation of vesicles (10.1073/pnas.1505405112).

Response: Thanks for the suggestion.

We are sorry for the misleading. Actually, the equation $[(A_0 - A_c)/A_0] \times 100\%$ depicts the ROS-responsive cleavage rate of peptide prodrug, rather than the degradation rate of crosslinked GelMA/5-FuA-Pep-MA hydrogels. In **Fig. 3g**, the ROS-responsive cleavage rate of 5-FuA-Pep-MA was characterized by HPLC, which was positively correlated with the H₂O₂ concentration and incubation time. Therefore, ROS levels in the pathological microenvironment of different stages of HS could change the response speed of prodrug. However, it does not directly correspond to the drug release of crosslinked GelMA/5-FuA-Pep-MA hydrogels. We have revised **Fig. 3g** and description to make it clear in the revised manuscript.

Before the drug release behavior of crosslinked GelMA/5-FuA-Pep-MA hydrogels was investigated, the released 5-FuA-Pro-Pro derived from ROS-cleaved 5-FuA-Pep-MA was verified by LC-MS (**Supplementary Fig. 9**). Then, the release curve of 5-FuA-Pro-Pro from crosslinked GelMA/5-FuA-Pep-MA hydrogels was plotted by HPLC with 5-FuA-Pro-Pro as a target molecule (**Fig. 3h**). Specifically, the drug release profile of crosslinked GelMA (0.2 g/mL) containing 5-FuA-Pep-MA (100 mM) was analyzed in PBS consisting of MMP9 (150 U/mL) and H₂O₂ (100 μM) at 37 °C. Specifically, the dried drug-loaded GelMA was dialyzed (1000 Da) in 5 mL of PBS containing MMP9 (150 U/mL) and H₂O₂ (100 μM). MMP9 was supplemented every

12 h to maintain enzyme activity. 50 μL of the solution was taken out at indicated time points, which was analyzed by HPLC (UltiMate 3000, ThermoFisher Scientific, USA) using the standard curve method. Each sample was measured in triplicate to calculate the mean and standard deviation. A sustained and approximately linear release of 5-FuA was obtained from the crosslinked GelMA/5-FuA-Pep-MA hydrogels with the UV-crosslinking time of 10 and 45 s. In addition, we also analyzed the drug release profile of crosslinked GelMA (0.2 g/mL) containing 5-Fu (100 mM) in PBS consisting of MMP9 (150 U/mL) and H_2O_2 (100 μM) at 37 $^\circ\text{C}$ using the same method. In comparison, the cumulative release of free 5-Fu in GelMA hydrogel crosslinked for 45 s was much higher (75%, Supplementary Fig. 10), indicating the burst drug release. We have added above description in Page 6 Line 38, Page 7 Line 18-19 and Page 25 Line 14-23 in the revised manuscript.

Indeed, we directly observe the drug release behavior of crosslinked GelMA/5-FuA-Pep-MA hydrogels using HPLC, and we believe that these data (**Supplementary Fig. 9** and **Fig. 3h**) are sufficient to reveal the drug release behavior of crosslinked MNs.

Fig. 3 Characterization, efficient drug loading and responsive drug release of MN patches. **a**, Force curve of MN patches with different crosslinking time. Arrows indicate fracture force. **b**, Swelling ratios of MNs with different crosslinking time. **c**, Drug loading amount of 5-FuA prepared from various concentrations of drug loading solutions. **d**, Crosslinking efficiency of MNs with different crosslinking time. **e**, Enzymatic degradation of GelMA with different crosslinking time. **f**, SEM images of MN patches before (left) and after (right) enzymatic degradation. **g**, ROS-responsive cleavage of 5-FuA-Pep-MA prodrug incubated with different concentrations of H_2O_2 . **h**, Drug release curves of dual responsiveness drug delivery platform with different crosslinking time. Error bars represent the standard deviation ($n = 3$). $***P < 0.001$.

Supplementary Fig. 9 Released fragments from MNs drug delivery platform. a, HPLC for drug release at different time points. **b,** ESI-HRMS (positive) spectrum of the effluent of HPLC at 18.7 min. **c,** ESI-HRMS (positive) spectrum of the effluent of the HPLC at 19.3 min.

Reviewers' Comments:

Reviewer #2:

Remarks to the Author:

All issues have been solved. Good to go.